

Climate
of the Past

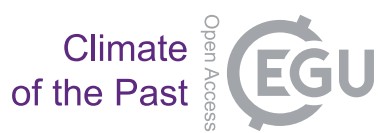

# Evolution of mean ocean temperature in Marine Isotope Stage 4

**Sarah Shackleton**[1,a]**, James A. Menking**[2]**, Edward Brook**[2]**, Christo Buizert**[2]**, Michael N. Dyonisius**[3,b]**,**
**Vasilii V. Petrenko**[3]**, Daniel Baggenstos**[4]**, and Jeffrey P. Severinghaus**[1]

[1]Scripps Institution of Oceanography, University of California, San Diego, La Jolla, 92093, USA
[2]College of Earth, Ocean, and Atmospheric Sciences, Oregon State University, Corvallis, 97331, USA
[3]Department of Earth and Environmental Sciences, University of Rochester, Rochester, 14642, USA
[4]Climate and Environmental Physics, Physics Institute and Oeschger Centre for Climate Change Research,
University of Bern, Bern, Switzerland
[a]present address: Department of Geosciences, Princeton University, Princeton, 08544, USA
[b]present address: Physics of Ice, Climate and Earth, Niels Bohr Institute, University of Copenhagen, Copenhagen, Denmark

**Correspondence:** Sarah Shackleton (ss77@princeton.edu)

**Abstract.** Deglaciations are characterized by relatively fast and near-synchronous changes in ice sheet volume, ocean temperature, and atmospheric greenhouse gas concentrations, but glacial inception occurs more gradually. Understanding the evolution of ice sheet, ocean, and atmosphere conditions from interglacial to glacial maximum provides insight into the interplay of these components of the climate system. Using noble gas measurements in ancient ice samples, we reconstruct mean ocean temperature (MOT) from 74 to 59.7 ka, covering the Marine Isotope Stage (MIS) 5a–4 boundary, MIS 4, and part of the MIS 4–3 transition. Comparing this MOT reconstruction to previously published MOT reconstructions from the last and penultimate deglaciation, we find that the majority of the last interglacial–glacial ocean cooling must have occurred within MIS 5. MOT reached equally cold conditions in MIS 4 as in MIS 2 ($-2.7 \pm 0.3\,°C$ relative to the Holocene, $-0.1 \pm 0.3\,°C$ relative to MIS 2). Using a carbon cycle model to quantify the $CO_2$ solubility pump, we show that ocean cooling can explain most of the $CO_2$ drawdown ($32 \pm 4$ of 40 ppm) across MIS 5. Comparing MOT to contemporaneous records of benthic $\delta^{18}O$, we find that ocean cooling can also explain the majority of the $\delta^{18}O$ increase across MIS 5 (0.7 ‰ of 1.3 ‰). The timing of ocean warming and cooling in the record and the comparison to coeval Antarctic isotope data suggest an intimate link between ocean heat content, Southern Hemisphere high-latitude climate, and ocean circulation on orbital and millennial timescales.

## 1 Introduction

The classical view of Pleistocene glacial cycles is a slow buildup of ice sheets followed by rapid disintegration (Abeouchi et al., 2013; Emiliani, 1955; Hays et al., 1976; Imbrie et al., 1993). However, the glacial inception – the transition from interglacial to glacial maximum – also involves global cooling, large-scale changes in ocean circulation, and carbon cycle reorganization that may not coincide with the gradual pacing of ice sheet growth. The last glacial inception was punctuated by a rapid global cooling 70 ka at the Marine Isotope Stage (MIS) 5a–4 boundary (Lisiecki and Raymo, 2005). During this period, nearly half of the interglacial–glacial drawdown of atmospheric $CO_2$ occurred over roughly 4 kyr (Ahn and Brook, 2008). This transition also brought extensive global cooling, buildup of polar ice sheets, and changes in deep ocean circulation (Adkins, 2013; Bereiter et al., 2012; Cutler et al., 2003; Yu et al., 2016). The mechanisms behind these rapid changes are not yet fully understood.

Multiple lines of oceanographic evidence (Adkins, 2013; Piotrowski, 2005; Thornalley et al., 2013; Yu et al., 2016) suggest that the MIS 5a–4 boundary marks the transition from the interglacial to glacial mode of ocean circulation. MIS 4 (like MIS 2) is characterized by cold conditions in both hemispheres and by the near absence of millennial-scale variability (Fig. 1). Sea surface temperature records for MIS 4 and MIS 2 (Kohfeld and Chase, 2017; Snyder, 2016) in-

dicate that these two intervals were comparably cold on the global scale, although the spatial distribution of temperature may have differed (Kohfeld and Chase, 2017). While similarities exist between the MIS 4 and MIS 2 intervals, there are notable differences. Atmospheric $CO_2$ was $\sim 20$ ppm lower in MIS 2 compared with MIS 4. Northern Hemisphere ice sheets and total ice volume were not as extensive as they were during MIS 2 (Cutler et al., 2003), but MIS 4 conditions included a greater extent of many glacier systems across the globe (Doughty et al., 2021; Schaefer et al., 2015). Understanding how and why conditions in MIS 4 and MIS 2 differed provides important context for the evolution of climate conditions during glacial inception.

One powerful indicator of global climate is the mean ocean temperature (MOT), which can be reconstructed using atmospheric noble gas ratios in ice-core-trapped air (Headly and Severinghaus, 2007). The total inventory of krypton and xenon in the ocean–atmosphere system is fixed, and the portion of the total that is dissolved in the global oceans depends on the MOT, as solubility of these heavy noble gases is strongly temperature dependent (Ritz et al., 2011). Ice-core-trapped air's $Kr/N_2$, $Xe/N_2$, and $Xe/Kr$ reflect the fraction of the noble gas inventory not dissolved in the ocean, which allows MOT at that time to be reconstructed. High-resolution reconstructions of MOT have been limited to the last two glacial terminations (Baggenstos et al., 2019; Bereiter et al., 2018a; Shackleton et al., 2019, 2020) but have provided unique insight into the interplay of key climate variables. In addition to the long-term warming across these deglaciations, millennial-scale variations in MOT are observed, which are also seen in Antarctic isotope records (Masson-Delmotte et al., 2010), and correspond to changes in Atlantic Meridional Overturning Circulation (AMOC) (Mcmanus et al., 2004). These deglacial features of MOT suggest an intriguing link between ocean circulation and ocean heat content. However, it is unclear if this link is unique to terminations or also applies to Dansgaard–Oeschger (DO) events (Dansgaard et al., 1982), which are millennial-scale climate oscillations that are thought to be linked to AMOC variability within glacial intervals (Lynch-Stieglitz, 2017; Stocker and Johnsen, 2003).

Here, we reconstruct MOT from 74 to 59.7 ka, covering the MIS 5a–4 transition, MIS 4, and part of the MIS 4–3 transition. The new record serves several purposes. First, it allows for a direct MIS 2–MIS 4 comparison, to assess their relative climate and ocean states. Second, comparison of MOT to benthic $\delta^{18}O$ changes from the onset of the last interglacial (MIS 5e) to MIS 4 and MIS 2 provides insight into the temporal evolution of ocean temperature and ice volume changes over the glacial cycle. Third, it allows us to test if the link between changes in ocean circulation and heat content exists during DO event 19 (DO19) at 72.1 ka. Last, using a simple carbon cycle model (Bauska et al., 2016), we estimate the contribution of whole-ocean cooling to the decrease in atmospheric $CO_2$ across MIS 5 and the MIS 5a–4 boundary due to the solubility pump.

## 2  Methods

### 2.1  Site description and ice core measurements

Ice samples were obtained by drilling a shallow (20 m, 0.24 m diameter) ice core at Taylor Glacier, Antarctica, a blue ice area located in the McMurdo Dry Valleys (Baggenstos et al., 2017). The core contains ice spanning gas ages from $\sim 58$ ka near the surface to 74 ka at 20 m depth (Menking et al., 2019). We excluded samples above 4 m depth to avoid alteration/contamination due to near-surface thermal fractures (Baggenstos et al., 2017). A total of 56 samples (including 11 replicate samples from identical depths) from Taylor Glacier were measured, with an average sample weight of 806 g and a mean record temporal resolution of 330 years. In addition, four WAIS (West Antarctic Ice Sheet) Divide samples from late MIS 4 ($\sim 66$–64 ka) were measured to replicate the Taylor Glacier results using samples from a different ice core. All ice core samples were analyzed for $Kr/N_2$, $Xe/N_2$, and $Xe/Kr$ using the method described by Bereiter et al. (2018b). The average of the three noble gas ratios was used to determine the final MOT following procedures in Shackleton et al. (2019). For brevity, we will refer to the MOT reconstructed from measured noble gases as "MOT data" in this work.

### 2.2  Taylor Glacier age model

We apply the ice core age model of Menking et al. (2019) with slight modifications for the MOT reconstruction. The age model was developed by matching measured variations in $CH_4$ and $\delta^{18}O_{atm}$ in the Taylor Glacier ice core to deep ice core records on the Antarctic Ice Core Chronology 2012 (AICC2012) timescale (Veres et al., 2013). Tie points were manually selected, and noble gas sample ages were determined from linear interpolation between tie points. For this study, we selected tie points from the higher-resolution North Greenland Ice Core Project (NGRIP; rather than EPICA Dronning Maud Land – EDML) $CH_4$ record on AICC2012, as well as three additional tie points from the EDML $CO_2$ record, also on AICC2012 (Table 1). Tie point uncertainties are reported relative to AICC2012 and do not include age uncertainty of the AICC2012 chronology itself. Tie point uncertainties have a minimal impact on the interpretation of the record.

### 2.3  Fractionation corrections of the noble gas ratios

The noble gas ratios measured in ice cores must be corrected for fractionation that occurs within the firn, which alters the noble gas ratios from their original atmospheric values (Headly and Severinghaus, 2007). We apply the correction approach of Shackleton et al. (2019), which uses a linear least-squares method to solve and correct for gravitational (Schwander, 1989) and thermal (Severinghaus et al., 1998)

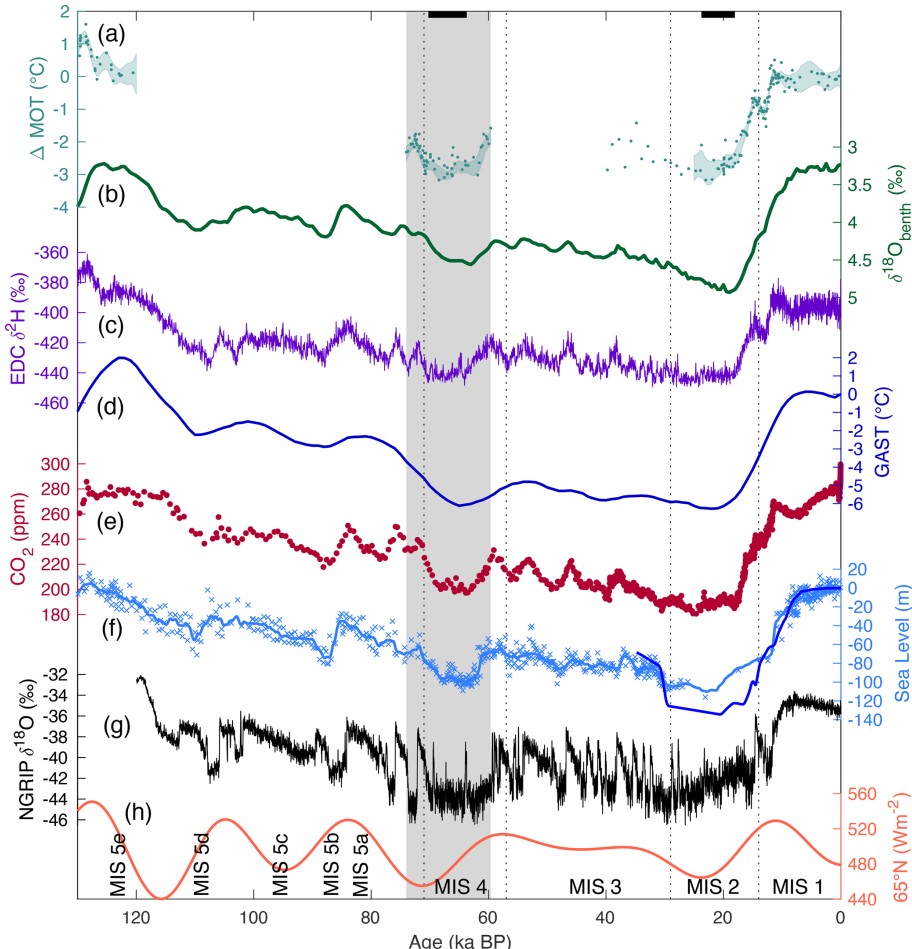

**Figure 1.** Climate records of the last glacial cycle. Panel **(a)** displays the mean ocean temperature (MOT) anomalies relative to the early Holocene (11–10 ka; Baggenstos et al., 2019; Bereiter et al., 2018a; Shackleton et al., 2019, 2020; this study). Shading shows the $1\sigma$ confidence envelope of MOT data from a spline with a 2500-year cutoff period and bootstrapping. For 0–25 ka, the spline includes all published records from this interval, and it ends at 25 ka to exclude EDC MOT data within the bubble–clathrate transition zone. Also shown are **(b)** the global benthic $\delta^{18}O$ stack (Lisiecki and Stern, 2016), **(c)** EDC $\delta^2H$ (Jouzel et al., 2007), **(d)** the global average surface temperature anomaly from present (Snyder, 2016), **(e)** the $CO_2$ composite record (Bereiter et al., 2015), **(f)** relative (light blue) (Grant et al., 2012) and eustatic (royal blue) (Lambeck et al., 2014) sea level, **(g)** NGRIP $\delta^{18}O$ (Andersen et al., 2004), and **(h)** summer solstice insolation at 65° N (Berger and Loutre, 1991). Dotted lines show boundaries between Marine Isotope Stages (MIS) from Lisiecki and Raymo (2005). Gray shading shows the interval of the mean ocean temperature (MOT) record presented in this study. Black bars at the top of the figure show the intervals used to define MIS 4 (this study) and MIS 2 (Bereiter et al., 2018a) MOT.

fractionations using measured isotope ratios of argon, nitrogen, and krypton. Further details on the applied fractionation corrections are included in Appendix A.

Corrections are more robust when calculating relative MOT change, rather than absolute MOT values, because errors in the fractionation corrections may produce a systematic offset in the corrected noble gas ratios, whereas the relative changes in these ratios are minimally influenced (Shackleton et al., 2020 and Appendix A). Therefore, we report MOT relative to Holocene MOT measured in the same ice core with the same applied method of fractionation correction. For the Taylor Glacier samples, we compare our data to five early Holocene (10.6 ka) replicate Taylor Glacier sam-

ples from Shackleton et al. (2020). WAIS Divide samples are reported relative to the average of Holocene samples from 11 to 10 ka ($n = 4$) (Bereiter et al., 2018a). WAIS Divide (Bereiter et al., 2018a) and EPICA Dome C (EDC) (Baggenstos et al., 2019) records both suggest that the entire Holocene was a very stable interval for MOT (respective $1\sigma$ standard deviations of 0.2 and 0.1 °C for all Holocene samples), so the particular choice of reference Holocene interval has minimal impact on the reported results.

**Table 1.** Tie points used in this study. Taylor Glacier $CH_4$, $\delta^{18}O_{atm}$, and $CO_2$ measurements are tied to preexisting records of $CH_4$ (Baumgartner et al., 2014), $\delta^{18}O_{atm}$ (Capron et al., 2010), and $CO_2$ (Bereiter et al., 2012) from well-dated ice cores on the AICC2012 (Veres et al., 2013) chronology.

| Gas age (ka) | Depth (m) | $Age_{min}$ (ka) | $Age_{max}$ (ka) | Data | Source |
|---|---|---|---|---|---|
| 59.02 | 3.15 | 58.7 | 59.2 | $CH_4$ | NGRIP |
| 59.77 | 4.19 | 59.68 | 59.97 | $CH_4$ | NGRIP |
| 60.45 | 5.125 | 59.8 | 62.5 | $CO_2$ | EDML |
| 63.72 | 7.2 | 62.6 | 64.18 | $CO_2$ | EDML |
| 64.2 | 7.79 | 63.86 | 64.5 | $CH_4$ | NGRIP |
| 70.35 | 11.5 | 69.2 | 70.94 | $CO_2$ | EDML |
| 71 | 13.25 | 70.43 | 71.95 | $CH_4$ | NGRIP |
| 72.34 | 16.2 | 72.15 | 72.64 | $CH_4$ | NGRIP |
| 72.7 | 17.4 | 72.2 | 73.3 | $\delta^{18}O_{atm}$ | NGRIP |
| 73.74 | 19.27 | 73.35 | 74.5 | $\delta^{18}O_{atm}$ | NGRIP |

## 2.4 MOT calculations from corrected noble gas ratios

We employ the box model of Bereiter et al. (2018a), which calculates the MOT anomaly relative to the modern ocean from the firn-corrected noble gas ratios. Parameterizations of the MOT box model applied in this study are detailed in Baggenstos et al. (2019). In addition, we use the recently published xenon and krypton solubilities from Jenkins et al. (2019). The box model requires input of sea level (Grant et al., 2012) to account for changes in the oceanic reservoir of xenon, krypton, and nitrogen that are unrelated to ocean temperature change. This includes changes in ocean volume, salinity, and sea surface pressure (Headly and Severinghaus, 2007). For Holocene MOT reference data and the MIS 2 data against which the record is compared, we use the sea level record of Lambeck et al. (2014) in the MOT box model. We also reevaluate the WAIS Divide Holocene and MIS 2 MOT record (Bereiter et al., 2018a), applying the same box model parameterizations as applied in this study (and in Shackleton et al., 2020) for a consistent comparison.

## 2.5 Carbon cycle model calculations of the solubility pump

To estimate the effect of a cooling ocean on the atmospheric $CO_2$ concentration via the solubility pump, we use a simple carbon cycle model (Bauska et al., 2016) to run a forward scenario of prescribed ocean temperature change from MOT constraints. The model consists of 14 boxes, representing the surface oceans (6 boxes), intermediate oceans (2 boxes), and deep oceans (3 boxes), a well-mixed atmosphere (1 box), and a terrestrial biosphere (2 boxes). The model simulates thermohaline circulation and mixing, air–sea gas exchange, export production, sediment burial/$CaCO_3$ compensation, and exchange of carbon between the atmosphere and terrestrial biosphere. More details on the model can be found in Appendix B.

## 2.6 Error analysis

The error on our MOT reconstruction is estimated by propagating all known uncertainties with 10 000 Monte Carlo simulations of our data. Sources of uncertainty include the analytical uncertainties for the noble gas ratios as well as the isotope ratios used for firn fractionation corrections. Additional uncertainties include the age uncertainty for the Taylor Glacier tie points (Table 1) and temporal and analytical uncertainties in the sea level curve. The calculated uncertainty for an individual sample is $0.2\,°C$ $(1\sigma)$, and the pooled standard deviation of the 11 replicate samples is $0.3\,°C$. Systematic errors (such as changes in ocean saturation state) may cause us to underestimate total uncertainty in our record.

To produce the splined MOT record, we use the 10 000 Monte Carlo iterations of the dataset and randomly sample the 56 individual MOT points via bootstrapping using the "randsample" MATLAB function with replacement. We then fit each of the 10 000 time series using a spline with a 2500-year cut-off period using the "csaps" MATLAB function and average the resulting splines to produce a final, smoothed version of our MOT record including uncertainty estimates.

## 3 Results

### 3.1 MIS 5a–4 boundary

During the rapid drawdown in atmospheric $CO_2$ ($\sim$ 72–68 ka, Fig. 2), we observe mean ocean cooling in two phases, with an overall net cooling of $0.9 \pm 0.3\,°C$ $(1\sigma)$. In the first phase (72–70 ka), MOT decreased by $0.7 \pm 0.3\,°C$ over roughly 2 kyr, coincident with Antarctic cooling and Greenland Interstadial 19 (GI19). In the second phase (70–68 ka), MOT stabilized and then decreased by a further $0.2 \pm 0.3\,°C$, reaching a minimum around 67.5 ka.

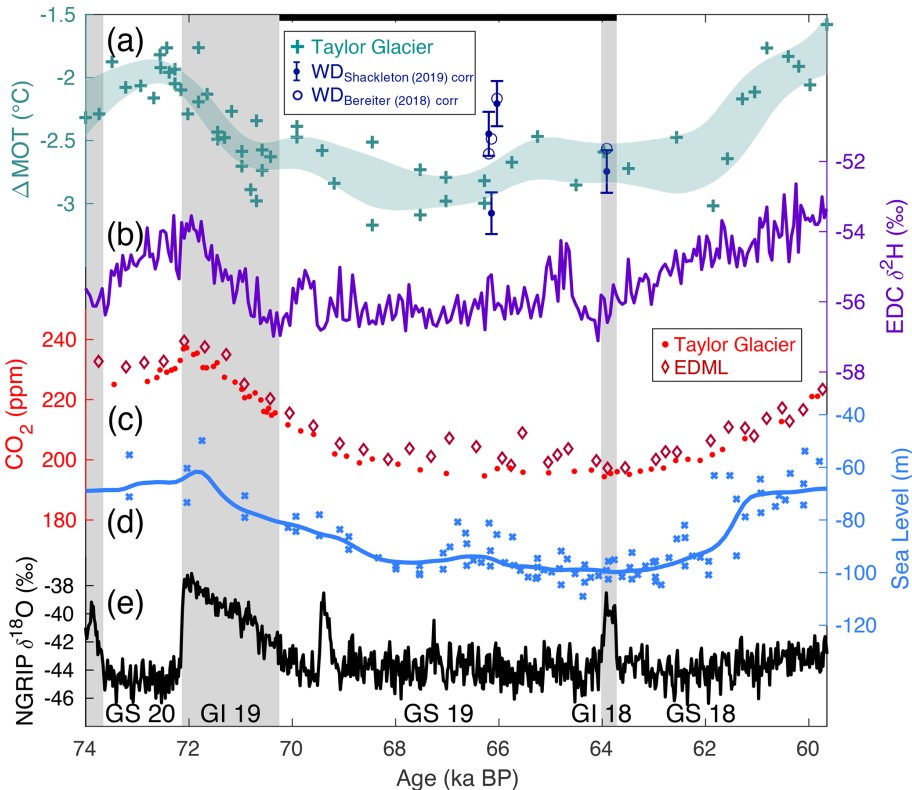

**Figure 2.** Mean ocean temperature (MOT) anomalies relative to the Holocene versus key climate variables. Panel **(a)** shows MOT data from Taylor Glacier (turquoise) and WAIS Divide (blue). Crosses indicate individual Taylor Glacier MOT data, and shading shows the $1\sigma$ confidence envelope of the Taylor Glacier data from a spline with a 2500-year cutoff period and bootstrapping. Solid blue points show WAIS Divide data corrected as described in Sect. 2 (with $1\sigma$ error bars), and open circles show the MOT results if the firn corrections detailed in Bereiter et al. (2018a) are applied. Also shown are **(b)** EDC $\delta^2$H (Jouzel et al., 2007) corrected for changes in seawater $\delta^2$H (see Appendix B), **(c)** $CO_2$ from EDML (diamonds) (Bereiter et al., 2012) and Taylor Glacier (points) (Menking et al., 2019) on AICC2012, **(d)** the relative sea level record (Grant et al., 2012), and **(e)** NGRIP $\delta^{18}O_{ice}$ (Andersen et al., 2004) on AICC2012. Gray panels show warm Greenland intervals (interstadials), and white panels indicate cold Greenland intervals (stadials). The black bar at the top of the figure shows the time interval used to calculate Marine Isotope Stage 4 MOT.

## 3.2 Comparison of MOT in MIS 4 and MIS 2

Here, we do not use the intervals identified and defined by benthic $\delta^{18}$O to compare MOT in MIS 4 and MIS 2, as the alignment of ice core and sediment records is uncertain, particularly in MIS 4. Instead, we define MIS 4 as the interval in which $CO_2$ and Antarctic temperature remain low and stable (70.3–63.7 ka, or Greenland Stadial 19 and Greenland Interstadial 18). For Taylor Glacier samples, we compare MIS 4 samples to five replicate MOT samples from MIS 2 (19.9 ka). For WAIS Divide samples, we compare the measured MIS 4 samples to all available, previously published (Bereiter et al., 2018a) MOT data from MIS 2 (24–18 ka) with the fractionation corrections and MOT box model parameterizations used in this study applied. The difference in WAIS Divide MOT results for the full MIS 2 interval ($n = 11$) versus 20–19 ka ($n = 4$) is less than 0.01 °C, so the difference in the selected intervals to define MIS 2 for each core should not affect the MIS 4–2 comparison.

The Taylor Glacier data show that MIS 4 MOT was statistically indistinguishable from MIS 2 ($-0.1 \pm 0.3$ °C relative to MIS 2 or $-2.7 \pm 0.3$ °C relative to the Holocene). The four WAIS Divide MIS 4 samples cover a narrower interval (66.2–63.9 ka) but are consistent with the Taylor Glacier results ($+0.1 \pm 0.5$ °C relative to MIS 2 or $-2.6 \pm 0.5$ °C relative to the Holocene). However, the WAIS data show more scatter. If we instead correct the WAIS Divide data for thermal fractionation using a firn model (Buizert et al., 2015), as in Bereiter et al. (2018a), and compare the results to MIS 2 data using this method of fractionation correction, we find that the WAIS Divide MIS 4 data are slightly less scattered (Fig. 2). With these corrections applied, the MIS 4 interval in WAIS Divide is 0.2 °C warmer than in MIS 2 ($-2.5 \pm 0.3$ °C relative to the Holocene). While all results are indistinguishable within error, they emphasize the importance of future work developing further understanding of firn air processes and their influence on MOT results. A detailed discussion of

the choice in fractionation corrections and their effect on calculated MOT is included in Appendix A.

### 3.3 MIS 4–3 transition

While our record may not capture the full transition into MIS 3, we find that there was substantial MOT warming towards the end of MIS 4. By 59.7 ka, MOT had reached levels comparable to the MOT peak at the end of MIS 5a at 72 ka (Fig. 2). Because our record does not contain a clear leveling of MOT, it is uncertain if or by how much MIS 3 MOT exceeded levels found at the end MIS 5a.

## 4 Discussion

### 4.1 Coevolution of MOT, benthic $\delta^{18}O$, $CO_2$, and Antarctic temperature during the last glacial inception

While the MOT proxy was developed over a decade ago (Headly and Severinghaus, 2007), only in the last few years have high-resolution MOT records become available (Baggenstos et al., 2019; Bereiter et al., 2018a; Shackleton et al., 2019, 2020). With the additional data from this study, we take the opportunity to review available MOT records and their relation to other key climate variables with a particular emphasis on the glacial inception (Fig. 3). While the available MOT records do not cover the entire last glacial cycle, we may still gain insight into climate evolution during (de)glaciations by comparing contemporaneous MOT, benthic $\delta^{18}O$, $CO_2$, and Antarctic $\delta^2 H$.

### 4.1.1 Evolving control of ocean temperature and ice sheet volume on benthic $\delta^{18}O$

The link between ocean temperature and benthic foraminiferal $\delta^{18}O$ (Fig. 3a) has long been recognized (Emiliani, 1955; Shackleton, 1974). While MOT represents volume-averaged ocean temperature, the intermediate and deep ocean make up the majority of total ocean volume. The benthic $\delta^{18}O$ record (Lisiecki and Stern, 2016) shown in Figs. 1 and 3 contains stacked records from intermediate and deep sites, and (when binned into ocean regions) covers approximately 70 % of the total ocean volume. Thus, changes in MOT should be largely reflected in temperature-driven changes in this $\delta^{18}O$ record. The scaling between MOT and $\delta^{18}O$ for ocean temperature change at 3.5 °C (Holocene/modern MOT, or $\Delta MOT = 0$) from Shackleton (1974) (0.26 ‰ °C$^{-1}$) is denoted by the gray arrow in Fig. 3a. While the temperature dependence of $\delta^{18}O$ from Shackleton (1974) is quadratic, it is effectively linear in the temperature range of the plotted MOT data, i.e., $d\delta^{18}O / dT$ varies by less than 6 % within the $\Delta MOT$ range shown in Fig. 3.

The other primary control on benthic $\delta^{18}O$ is ice volume. Considering the temporal evolution of $\delta^{18}O$ and MOT, it is possible to gain insight into the relative controls on $\delta^{18}O$ during the intervals where $\delta^{18}O$ and MOT data are available. Applying the benthic $\delta^{18}O$ temperature sensitivity from Shackleton (1974), we find that the ocean temperature anomaly during MIS 4 accounts for 0.7 ‰ of the 1.3 ‰ $\delta^{18}O$ anomaly relative to Holocene/modern benthic $\delta^{18}O$, implying that the remaining 0.6 ‰ is due to enhanced ice sheet volume. For comparison, the MIS 2–Holocene benthic $\delta^{18}O$ change is 1.7 ‰. Considering the MOT–$\delta^{18}O$ relationship for late MIS 5a/MIS 4 (light green in Fig. 3), late MIS 3 (cyan), and MIS 2 (light blue), there is some variability in MOT within these intervals, but average MOT across the intervals remains essentially unchanged. However, there is a clear, long-term increase in $\delta^{18}O$ across these intervals. The similarity in MOT between MIS 4 and MIS 2 suggests that the more positive benthic $\delta^{18}O$ during the latter stage is caused by a greater global ice volume. Taken together, these observations are consistent with previous studies (Cutler et al., 2003; Shakun et al., 2015; Waelbroeck et al., 2002) suggesting that ocean cooling outpaced Northern Hemisphere ice sheet growth in the last glacial inception. This decoupling of ocean cooling and ice sheet growth may be an important clue for future investigation of the mechanism of glacial cycles.

### 4.1.2 Early role of ocean cooling in atmospheric $CO_2$ drawdown

Here, we discuss two separate drawdowns of $CO_2$ during the glacial inception, each of which was approximately 40 ppm. The first occurred from MIS 5e to MIS 5a, and the second took place from MIS 5a to MIS 4. Using a carbon cycle model (Bauska et al., 2016), we estimate that the $3.1 \pm 0.4$ °C MOT decrease between the onset of MIS 5e (129 ka) and the end of MIS 5a (72 ka) accounted for $32 \pm 4$ of the $\sim 40$ ppm $CO_2$ reduction that occurred across this interval. We emphasize that the available MOT data spans 9 kyr at the onset and 2 kyr at the end of the long ($\sim 57$ kyr) MIS 5 interval, so our insight into the role of the solubility pump on $CO_2$ variations within MIS 5 is limited. However, our MOT data suggest a dominant role of ocean cooling on the $\sim 40$ ppm $CO_2$ drawdown that occurred across MIS 5e–5a. Much of this drawdown is focused on the MIS 5e–5d transition around 115 ka, making this period a high priority for future detailed ice core MOT reconstruction.

During the MIS 5a–4 transition, we estimate that the reconstructed MOT net decrease of 0.9 °C led to a $CO_2$ drawdown of $9 \pm 3$ ppm by solubility, which is a relatively small but not insignificant fraction of the $\sim 40$ ppm drawdown that occurred over the full interval (Fig. 4). A comparison of the Taylor Glacier records of MOT and $CO_2$ over the MIS 5a–4 transition ($\sim 72$–68 ka) suggests that while both MOT and $CO_2$ decreased during this transition, the overall trends appear distinct in their shapes. While the rate of $CO_2$ de-

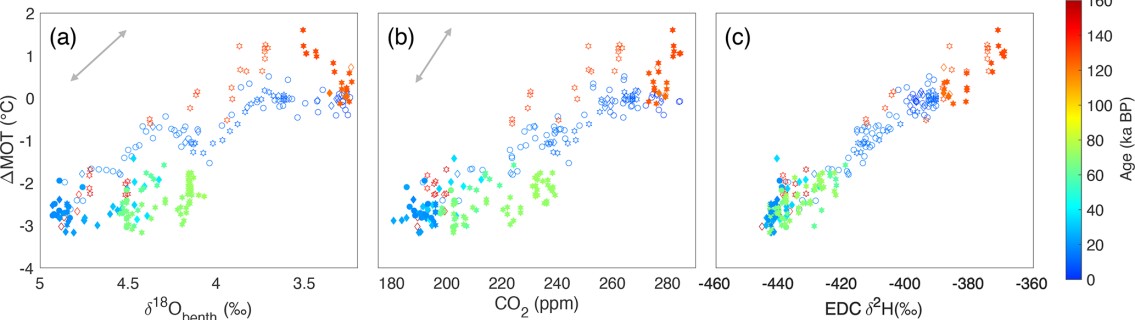

**Figure 3.** $\Delta$MOT plotted against coeval **(a)** $\delta^{18}O_{benth}$ (Lisiecki and Stern, 2016), **(b)** atmospheric $CO_2$ (Bereiter et al., 2015), and **(c)** EDC $\delta^2H$ (Jouzel et al., 2007) corrected for changes in seawater $\delta^2H$. The $\delta^{18}O_{benth}$, $CO_2$, and EDC $\delta^2H$ records were linearly interpolated in order to plot them against contemporaneous MOT. Additionally, the EDC $\delta^2H$ was smoothed using a Gaussian filter with a 500-year window to remove high-frequency variability. The gray arrow in panel **(c)** shows the $\Delta$MOT–$\delta^{18}O$ scaling from (Shackleton, 1974). The gray arrow in panel **(b)** shows the $\Delta$MOT–$CO_2$ relationship for the solubility pump from the carbon cycle model. Filled symbols include data from the last interglacial through glacial maximum (129–18 ka) to highlight the glacial inception, while open symbols indicate data from the last and penultimate deglaciations and the Holocene. Diamonds indicate MOT data constructed from the EDC record (Baggenstos et al., 2019; Shackleton et al., 2020), circles show data from WAIS Divide (Bereiter et al., 2018a, and this study), and stars show MOT data from Taylor Glacier (Shackleton et al., 2019, 2020, and this study). The color of data indicates the age.

crease was relatively constant over the full transition, MOT decreases more rapidly during the first half of the transition ($-0.41 \pm 0.09\,^\circ\text{C kyr}^{-1}$ over GI19, 72.1–70.3 ka) than in the second half ($-0.19 \pm 0.07\,^\circ\text{C kyr}^{-1}$, 70.3–67.5 ka). This duality in trends over the MIS 5a–4 transition has been observed in other proxy records (Barker and Diz, 2014) and may provide important insight into the evolving controls on atmospheric $CO_2$ during this interval. While ocean cooling may explain roughly one-third of the $CO_2$ drawdown in the first half of the transition, significant carbon cycle reorganization is required to explain the majority of the atmospheric $CO_2$ decrease, particularly in the second half of the transition.

Again, we emphasize that available MOT records do not cover the full glacial cycle, and substantial gaps in the data exist for MIS 5 and MIS 3. However, it is notable that the coevolution of $CO_2$ and MOT over the last glacial inception shows a strikingly similar trend to that of benthic $\delta^{18}O$ and MOT (Fig. 3); ocean cooling appears to play a dominant role in the net $CO_2$ decrease across MIS 5, but the long-term trend of $CO_2$ drawdown across MIS 4–2 does not correspond with ocean cooling, as MOT had reached levels comparable to MIS 2 by MIS 4. We speculate that, within the last glacial inception, the MIS 5a–4 boundary marks a distinct decoupling in MOT and $CO_2$ trends. This is consistent with the hypothesis presented by Adkins (2013): that the MIS 5a–4 boundary marks a transition between interglacial and glacial modes of ocean circulation and shifts the controls on carbon uptake from primarily temperature-driven solubility to circulation-driven storage, for example, via reduced ventilation of abyssal waters that allows respired carbon to accumulate there.

Our data allow us to put new constraints on the role of the solubility pump in atmospheric $CO_2$ variations across the studied intervals. Out of the full $\sim 80\,\text{ppm}$ $CO_2$ from MIS 5e to MIS 4, our modeling suggests that MOT changes can explain $41 \pm 4\,\text{ppm}$. These estimates of the solubility pump agree well with the canonical $10\,\text{ppm}\,^\circ\text{C}^{-1}$, (Williams and Follows, 2011). However, our MOT data provide no information on the spatial distribution of ocean temperature change, which a recent study suggests plays an important role in modulating the strength of the solubility pump via changes in ocean saturation state (Khatiwala et al., 2019). The referenced study found that changes in air–sea disequilibrium between interglacial and glacial ocean conditions enhanced the solubility pump by $\sim 60\,\%$ during the last glacial maximum. If such disequilibrium effects are also relevant for the timescales and periods considered here, our solubility-driven estimates from the carbon cycle model simulations may be considered a lower bound. In particular, if the enhanced disequilibrium effect is linked to the onset of the glacial mode of ocean circulation at the MIS 5a–4 transition, the solubility pump may play a larger role there than suggested from our simplified carbon cycle modeling.

We note that changes in ocean saturation state may also influence the noble gases and, thus, MOT estimates. However, the more rapid equilibration timescales of the noble gases versus $CO_2$ (a few weeks versus roughly a year) means that they are quite insensitive to disequilibrium (Ritz et al., 2011). However, given the recent improvements in analytical precision of the MOT technique, glacial–interglacial changes in noble gas disequilibrium merit future investigation.

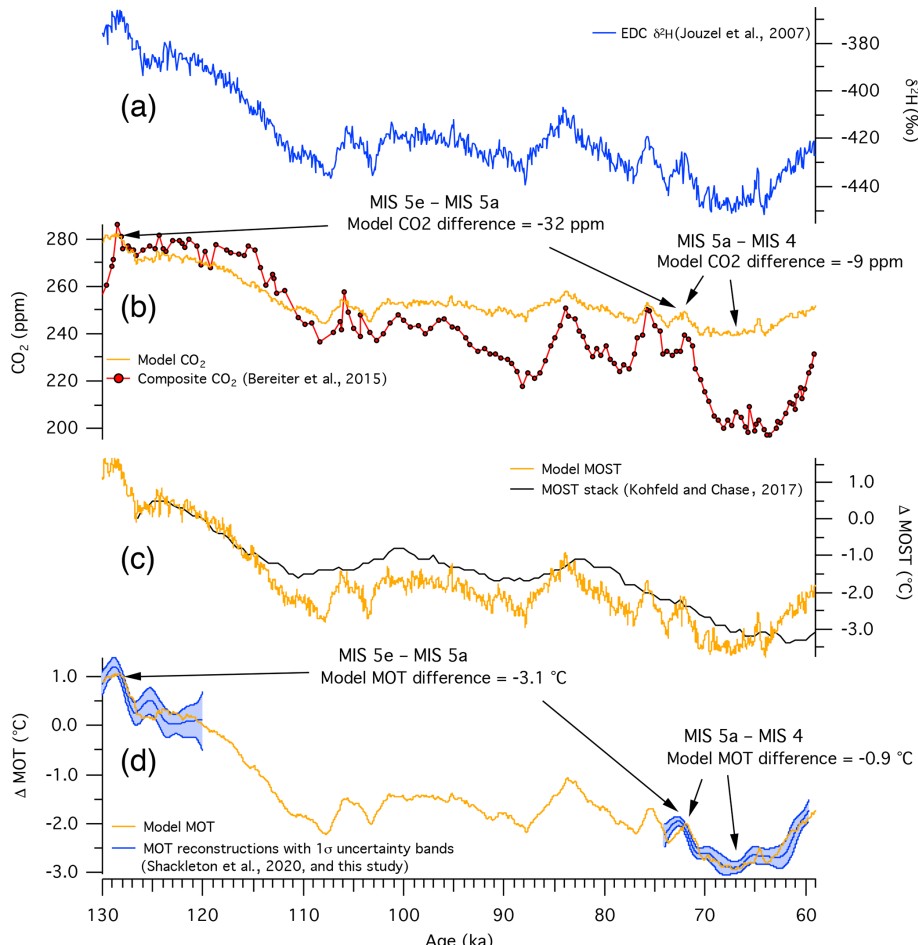

**Figure 4.** Results from a carbon cycle model estimating the magnitude of $CO_2$ drawdown due to mean ocean temperature cooling. Inputs to the model sea surface temperature changes (MOST, orange trace, **c**) are scaled to ice core $\delta^2H$ data corrected for seawater $\delta^2H$ changes (blue trace, **a**, see Appendix B). The sea surface temperature changes are transmitted to the deep ocean via circulation and mixing in the model, causing the mean ocean temperature (MOT, orange trace, **d**) to evolve through time. The modeled MOT history agrees well with the existing (but limited) ice core MOT data (blue traces, **d**). Ocean salinity also evolves in the model and is scaled to sea level data (Appendix B). The evolution of $CO_2$ in the model (orange trace, **b**) is only due to changes in ocean solubility. The modeled $CO_2$ history is compared to ice core $CO_2$ records (red markers) (Bereiter et al., 2015). Model results within 120–74 ka should be interpreted with caution, as MOT data do not exist for validation.

### 4.1.3    Strong correlation between MOT and Antarctic climate on orbital and millennial timescales

As highlighted in this and several other MOT studies (Bereiter et al., 2018a; Shackleton et al., 2019, 2020), one of the most striking features of MOT records is their strong correlation to Antarctic water isotope records (Fig. 3c). For the MOT data from this study, we find a lower correlation between MOT and EDC $\delta^2H$ ($n = 56$, $r^2 = 0.59$) than between all available MOT records ($n = 243$, $r^2 = 0.94$). However, MOT and $\delta^2H$ data for this interval cover a relatively narrow range compared with other records, resulting in a lower signal-to-noise ratio, and thus may explain the lower correlation. To test this hypothesis, we use the pooled standard deviation of replicate MOT samples (0.3 °C) as a predictor

of random noise in the MOT record to estimate the expected correlation between $\delta^2H$ and MOT if we assume they are perfectly correlated ($r^2 = 1$). Under these assumptions, we would predict $r^2$ values of $0.58 \pm 0.09$ and $0.93 \pm 0.01$ for the MIS 4 subsamples and all MOT samples, respectively, which is consistent with the observed values. It is remarkable that the MOT–$\delta^2H$ scaling is similar on millennial and orbital timescales, given that climate dynamics on these two timescales are likely to be different. Multiple explanations may be given for the strong correlation.

If there is indeed a causal relationship between MOT and Antarctic temperature, causality could plausibly run in either direction. First, it has been suggested that Southern Hemisphere high-latitude temperature, for which Antarctic $\delta^2H$ is a proxy, provides a control on MOT (Bereiter et al., 2018a).

Clim. Past, 17, 1–17, 2021                                       https://doi.org/10.5194/cp-17-1-2021

Given that a large fraction of the global ocean interior is ventilated in the Southern Ocean (Johnson, 2008), processes acting in the Southern Ocean around Antarctica are likely to be important in setting the MOT. The temperature of deep waters formed in the Southern Ocean, as well as the rate at which they form, is probably linked to Southern Hemisphere high-latitude climate, providing a pathway to control MOT variations (Bereiter et al., 2018a).

Second, it is possible that causality runs in the opposite direction, with MOT being a strong control on Antarctic $\delta^2$H. In their modeling study, Pedro et al. (2018) proposed a mechanism linking MOT to Antarctic temperature on millennial timescales, as part of their effort to provide a more thorough dynamical framework for the bipolar seesaw. Briefly, during weakened AMOC intervals, ocean warming centered in the intermediate-depth North Atlantic is spread throughout the ocean basins via Kelvin and Rossby waves, which cannot cross the Antarctic Circumpolar Current. The enhanced temperature gradient across the Antarctic Circumpolar Current drives poleward ocean and atmospheric eddy heat fluxes, which are amplified by sea ice reduction and the ice–albedo feedback. The net result is a strong warming of the Antarctic continent. In this view, it is feasible that the MOT controls Antarctic temperature, via variations in Southern Ocean poleward eddy heat transport and sea ice feedbacks.

Finally, MOT and Antarctic temperature need not be causally linked; the tight correlation between them may reflect a shared dependence on a third variable that is most likely AMOC variability. It is well established that Antarctic temperature responds to AMOC variations via the bipolar seesaw mechanism (Stocker and Johnsen, 2003). Likewise, AMOC variations and their associated changes in oceanic heat loss to the Arctic atmosphere have been shown to influence MOT in model simulations (Galbraith et al., 2016; Pedro et al., 2018 TS1). Thus, it is conceivable that both variables respond to AMOC variations without the necessity for a direct causal link between them.

Here, we remain agnostic as to which of these three explanations is the correct one. Such a determination would require detailed modeling studies that are beyond the scope of the present work. However, our record demonstrates that Antarctic temperature and MOT covary on millennial timescales during DO19, suggesting that their link is not unique to deglaciations and is a general feature of the climate system.

### 4.2   The cold and stable MIS 4 interval

Given that global ice volume was greater (Cutler et al., 2003) and $CO_2$ concentrations were lower in MIS 2 than in MIS 4, the comparably cold conditions during these intervals suggested in this study and by recent (Doughty et al., 2021) and previous (Kohfeld and Chase, 2017; Snyder, 2016) work is somewhat puzzling. All else being equal, the 20 ppm higher atmospheric $CO_2$ in MIS 4 would lead to 0.5 °C warmer

global average surface temperatures than in MIS 2, assuming a climate sensitivity of 3.5 °C. Other forcing may be required to resolve this conundrum. While changes in planetary albedo are often assumed to scale with ice volume, this may not be appropriate when comparing MIS 4 and MIS 2. Doughty et al. (2021) suggest that while ice volume was lower, glacier extent may have been greater in MIS 4 than in MIS 2. This may have led to a higher planetary albedo in MIS 4 than in MIS 2. However, the authors suggest that the greater glacial extent was due to the cold conditions in MIS 4, so there is some circularity to this argument.

Another notable difference between in MIS 4 and MIS 2 exists for the Saharan region. Proxy records of the Sahara suggest that MIS 4 was a uniquely arid interval within the last glacial cycle (Castañeda et al., 2009; Skonieczny et al., 2019; Tierney et al., 2017), while regions of the eastern Sahara and Sinai Desert during MIS 2 may have been wetter than today (Hamdan and Brook, 2015), suggesting greener Saharan conditions in MIS 2 compared with MIS 4. Climate simulations of a greener Sahara suggest globally warmer temperatures due to lower albedo and higher atmospheric moisture (Tabor et al., 2020). We speculate that more arid Saharan conditions in MIS 4 may have contributed additional cooling of MIS 4 relative to MIS 2 and, in part, countered the warming effect of higher $CO_2$ during this interval. Of course, this new MOT record cannot shed light on the conditions of the low-latitude hydrosphere. However, it adds to a growing body of work suggesting that, despite the smaller ice volume and higher atmospheric $CO_2$, MIS 4 was comparably cold to MIS 2. We believe that this conundrum merits further investigation and may be a valuable target for forthcoming climate modeling efforts.

In addition to their cold temperatures, MIS 4 and MIS 2 also share an absence of millennial-scale variability. Theories explaining the apparent lack of bipolar seesaw behavior during very cold periods (such as MIS 4 and MIS 2) have invoked mechanisms related to thresholds in ice volume (McManus et al., 1999) and Southern Ocean temperature (Buizert and Schmittner, 2015). While ice volume during MIS 2 exceeds that of MIS 4, MOT during MIS 2 and MIS 4 indicate equally cold ocean conditions. This supports the idea that thresholds in ocean temperature, rather than global ice volume, may determine the presence or absence of millennial-scale variability within a glacial.

## 5   Conclusions and future outlook

Our record adds to the growing number of MOT reconstructions and provides unique insight into the climate conditions of MIS 4 within the context of the last glacial inception. The MOT record shows comparably cold and stable conditions in MIS 4 and MIS 2. As demonstrated in previous studies, MOT and Antarctic isotope records are remarkably correlated and covary on millennial timescales during DO19, pro-

viding the first evidence of the connection between MOT, Antarctic temperature, and inferred AMOC variability outside of deglaciations. Comparisons of coeval benthic $\delta^{18}O$, $CO_2$, and MOT show that while MOT reaches a minimum in the last glacial cycle by MIS 4, $CO_2$ and $\delta^{18}O$ do not achieve their glacial extrema until MIS 2; ocean cooling outpaced ice sheet growth and atmospheric $CO_2$ drawdown in the last glacial inception.

Using a carbon cycle model and ocean temperature constraints provided from our record, we demonstrate that ocean cooling played a major role in the early (MIS 5) stages of atmospheric $CO_2$ drawdown in the glacial inception (32 of $\sim 40\,ppm$), a moderate role in the first half of the MIS 5a–4 transition (7 of $\sim 20\,ppm$), a minor role in the second half of the 5a–4 transition (2 of $\sim 20\,ppm$), and no measurable role in the $\sim 20\,ppm$ decrease between MIS 4 and MIS 2. This suggests an evolving control on atmospheric $CO_2$ during the glacial inception in which the solubility pump initially dominates but plays a progressively smaller role as the glacial inception progresses. MOT reconstruction of the entire MIS 5 interval would provide valuable insight into this apparent trend.

Studies comparing the $CO_2$ (Bereiter et al., 2012) and Atlantic Western Boundary Undercurrent (Thornalley et al., 2013) response to millennial-scale variability during MIS 5 and MIS 3 suggest that the changes in ocean circulation at the MIS 5–4 boundary altered the nature of abrupt climate change between these two intervals. Comparison of the MOT response to DO cycles within MIS 5 and MIS 3 may prove useful in understanding the nature of this transition.

This study demonstrates that it is possible to capture MOT changes during the larger of the millennial-scale DO events using the noble gas ratio technique. However, comparison of the MOT records between smaller DO events will push the current analytical limits of this method. While improvements in analytical precision will benefit future studies, an improved understanding of gas fractionation processes within the ice and firn, and the mechanisms of air–sea gas exchange will be critical to accurate interpretation of ice core MOT records.

## Appendix A: Comparison of MOT results using different methods of fractionation corrections and between Kr/N₂, Xe/N₂, and Xe/Kr

Gases are trapped in bubbles in ice during the process of firnification, as snow compacts and densifies into firn and eventually glacial ice. This process is gradual, occurring on timescales on the order of hundreds to thousands of years. During this time, the low permeability of the firn restricts bulk air motion but allows for air in the open pores to exchange with the overlying atmosphere and throughout the firn column primarily through molecular diffusion. This mechanism of air transport allows for processes such as grav-

itational settling (Schwander, 1989), thermal diffusion (Severinghaus et al., 1998), and kinetic fractionation (Birner et al., 2018; Buizert and Severinghaus, 2016; Kawamura et al., 2013) to alter $Kr/N_2$, $Xe/N_2$, and $Xe/Kr$ from their atmospheric compositions before bubble close-off. Correction of $Kr/N_2$, $Xe/N_2$, and $Xe/Kr$ for these processes may be done with output from a firn model and/or from measurements of isotope ratios of inert gases (such as argon, nitrogen, krypton, and xenon), which are also influenced by these processes but are unchanging in the atmosphere. Argon isotope ratios are a slight exception, due to the gradual degassing of $^{40}Ar$ from the solid earth (Bender et al., 2008). With the known rate of change in atmospheric $^{40}Ar$ and age of the samples, a small ($<0.005\,‰$) correction is applied to measured $\delta^{40/38}Ar$ and $\delta^{40/36}Ar$. In the case of this study, we measure isotope ratios of argon ($\delta^{40/36}Ar$, $\delta^{40/38}Ar$), nitrogen ($\delta^{15}N$–$N_2$), and krypton ($\delta^{86/82}Kr$, $\delta^{86/83}Kr$, $\delta^{86/84}Kr$) for the applied fractionation corrections.

The approach of fractionation correction in this study differs slightly from that of Bereiter et al. (2018a), which uses previously published firn model output from Buizert et al. (2015) to correct $Kr/N_2$, $Xe/N_2$, and $Xe/Kr$ for thermal fractionation. To correct $Kr/N_2$, $Xe/N_2$, and $Xe/Kr$ for gravitational fractionation, the Bereiter et al. (2018a) study uses measured $\delta^{40/36}Ar$, which is first corrected for thermal fractionation using this same firn model output. While in the case of the WAIS Divide ice core, previously published firn model output was readily available for this purpose, no such model output exists for Taylor Glacier, which is why this method of fractionation correction was not considered in this study.

In this study, as in Shackleton et al. (2019), ), the corrections for gravitational and thermal fractionation are done with measured inert gas isotope ratios. The reason for this choice over other considered methods (see below) is that (1) it gives the best agreement in Taylor Glacier MOT results between replicate samples for the Holocene and MIS 2 (Shackleton et al., 2020), and (2) it gives the best results for calculated MOT in firn air and surface ice samples from a wide range of site conditions (Shackleton, 2019). The average magnitude of the fractionation corrections for $\delta Kr/N_2$, $\delta Xe/N_2$, and $\delta Xe/Kr$ were 5.0‰, 8.9‰, and 3.9‰, respectively, for the Taylor Glacier samples in this study. The fractionation-corrected, atmospheric values found for $\delta Kr/N_2$, $\delta Xe/N_2$, and $\delta Xe/Kr$ were $-1.4‰$, $-3.6‰$, and $-2.1‰$, respectively. While the fractionation corrections are large relative to the atmospheric signal, they are dominated by gravitational fractionation, which is physically well understood and constrained by the measured isotope ratios. For context, the average magnitude of fractionation correction for the WAIS Divide samples in this study are 22.7‰, 42.5‰, and 19.3‰ for $\delta Kr/N_2$, $\delta Xe/N_2$, and $\delta Xe/Kr$, respectively. The large difference in the magnitude of the applied fractionation correction between ice cores but good agreement in corrected noble gas ratios and resulting MOT

gives us some confidence in our ability to apply these corrections.

To assess how differing methods of fractionation correction may impact the MOT results, we apply multiple corrections following Shackleton et al. (2020). Figure A1 shows the MIS 4 Taylor Glacier MOT data (average of the $Kr/N_2$, $Xe/N_2$, and $Xe/Kr$ results) with fractionation corrections for gravitational fractionation, gravitational and thermal fractionation, and gravitational and kinetic fractionation. For a detailed explanation of these methods of fractionation correction, see the supporting information of Shackleton et al. (2020). Results are compared between these differing methods of fractionation correction when (i) they are not calculated relative to a reference interval, (ii) they are calculated relative to Holocene MOT, and (iii) they are calculated relative to MIS 2 MOT. As previously shown, MOT reported relative to a reference interval in the same core is more robust to the method of fractionation correction than when no reference interval is used (Shackleton et al., 2020). However, even when the MIS 4 MOT data are referenced to Holocene MOT, there appears to be a small but systematic offset in the MOT results using different methods of fractionation correction. If the MIS 4 data are calculated relative to MIS 2 data from the same ice core, the offset is reduced.

Comparison of the MOT results from $Kr/N_2$, $Xe/N_2$, and $Xe/Kr$ when normalized to Holocene versus MIS 2 MOT data show a similar phenomenon to the observed offset in results between differing fractionation corrections (Fig. A2). The offset between the MOT results for the three noble gas ratios is present, regardless of the fractionation correction applied; if the MOT results from the individual ratios are reported relative to those from the Holocene, there is a small offset between the MOT results from $Kr/N_2$, $Xe/N_2$, and $Xe/Kr$. However, if the MOT difference is calculated between MIS 2 and our MIS 4 data, the offset diminishes.

The observed patterns are consistent with systematic uncertainties in the fractionation corrections applied to the noble gas ratios. Differences in site conditions, such as temperature, accumulation, and atmospheric circulation, can lead to differences in firn column height, temperature profile, and dynamics of gas transport and mixing within the firn. These have implications for gravitational, thermal, and kinetic fractionation of $Kr/N_2$, $Xe/N_2$, and $Xe/Kr$. While the fractionation corrections should account and correct for these changes, a systematic error in these corrections or the presence of an additional fractionating process that has not been accounted for may result in systematic error that varies with site condition. This would result in a similar magnitude of systematic error under similar firn conditions. Thus, the systematic differences in MOT results using different fractionation correction methods or between the three noble gas ratios are largest (up to $0.3 \pm 0.3\,°C$ between correction methods and $0.4 \pm 0.3\,°C$ between individual noble gas ratios) when comparing results between glacial and interglacial intervals but are minimal (up to $0.1 \pm 0.3\,°C$ between correc-

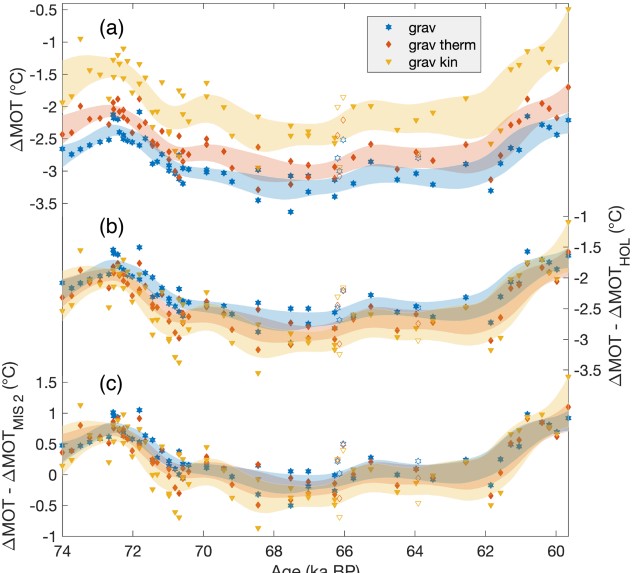

**Figure A1.** Comparison of mean ocean temperature (MOT) anomalies between three methods of fractionation correction. Results are for the average of the three MOT proxies ($Kr/N_2$, $Xe/N_2$, and $Xe/Kr$). Panel (**a**) shows results if the noble gas ratios are corrected for fractionation and no reference interval is used. Panel (**b**) shows the results when the noble gas ratios are reported relative to Holocene data, using the same method of fractionation correction. Panel (**c**) is the same as panel (**b**) but MOT is reported relative to MIS 2. Individual MOT data from Taylor Glacier are shown as filled symbols and the $1\sigma$ confidence envelope from a spline with a 2500-year cutoff period, and bootstrapping is shown using shading. WAIS Divide data are shown as open symbols.

tion methods and $<0.01 \pm 0.3\,°C$ between noble gas ratios) when comparing MOT results between the MIS 4 and MIS 2 glacial intervals.

However, a systematic error in fractionation correction may not be the only explanation for the offset in MOT results between $Kr/N_2$, $Xe/N_2$, and $Xe/Kr$. Processes that decouple atmospheric noble gas exchange from ocean heat exchange may also introduce systematic error in MOT reconstructions and may affect the krypton, xenon, and nitrogen to different degrees, resulting in differences in MOT reconstructed from $Kr/N_2$, $Xe/N_2$, and $Xe/Kr$. If this were the cause of the observed offset in MOT results between the three noble gas ratios, we would predict that the offset would be consistent between ice cores.

Considering the MIS 4, MIS 2, and Holocene data from the WAIS Divide record, the relatively sparse and somewhat noisier data make it difficult to discern any trends. However, if anything, the relative offset in the three noble gas proxies is the opposite of that observed for Taylor Glacier. This suggests that the primary mechanism to explain the observed differences in the MOT results between $Kr/N_2$, $Xe/N_2$, and $Xe/Kr$ is a process that affects these ratios within the firn or ice, rather than the atmospheric inventories of Xe, Kr, and

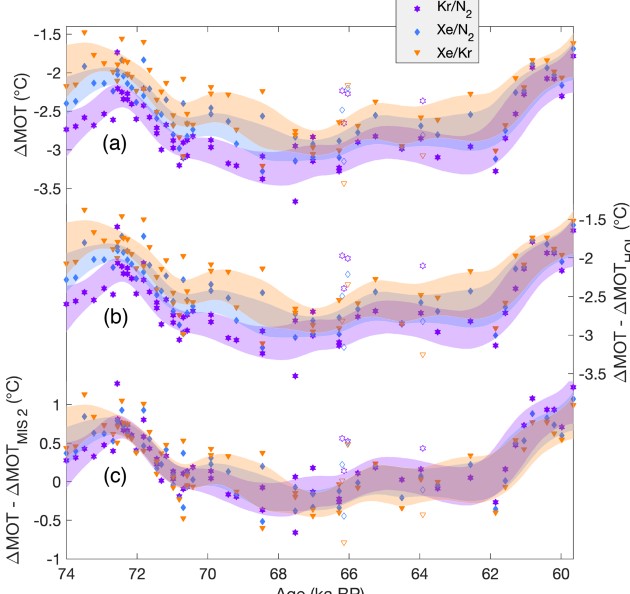

**Figure A2.** Mean ocean temperature (MOT) anomalies calculated from $Kr/N_2$, $Xe/N_2$, and $Xe/Kr$. Noble gas ratios are corrected for gravitational and thermal fractionation (Shackleton et al., 2019, and this study). Panel **(a)** shows the calculated MOT anomaly when no reference interval is used. Panel **(b)** shows the MOT anomaly relative to Holocene MOT results. Panel **(c)** shows the MOT anomaly relative to MIS 2. Individual MOT data from Taylor Glacier are shown using filled symbols and the $1\sigma$ confidence envelope from a spline with a 2500-year cutoff period, and bootstrapping is shown using shading. WAIS Divide data are shown using open symbols.

$N_2$. However, this does not rule out the existence of processes related to the latter. While the slight differences in results with different fractionation correction and between the three noble gas ratios do not affect the conclusions of the study, further investigation is necessary to gain a better understanding of these processes' influence on the MOT proxies and their associated uncertainties.

## Appendix B: Ocean solubility effect on $CO_2$

In order to estimate the magnitude of $CO_2$ drawdown between 129 and 59.7 ka due to a cooling ocean, we used a simple carbon cycle model and held all model parameters constant except for ocean surface temperatures. The model communicates ocean surface temperature changes to the deep ocean boxes through circulation and mixing; thus, surface temperature changes alter the MOT. The model spinup reached equilibrium with an atmospheric $CO_2$ value of 282 ppm, MOT of 4.5 °C, and a mean ocean surface temperature of 18 °C. Ocean surface temperatures in the six surface boxes were then allowed to vary such that the timing and relative magnitude of temperature in each box changed according to a linear scaling to the EPICA Dome C $\delta^2H$ record

(Jouzel et al., 2007) (Fig. 4). The $\delta^2H$ record was first corrected for changes in seawater $\delta^2H$ using the $\delta^{18}O$ seawater reconstruction of Waelbroeck et al. (2002) and 8 : 1 scaling of $\delta^2H : {}^{18}O$ changes. Whole ocean salinity change was also prescribed in the model to account for the solubility effect on $CO_2$. Salinity was linearly scaled to the Grant et al. (2014) sea level record assuming a preindustrial salinity of 34.72 psu (practical salinity units) and a Last Glacial Maximum salinity of 35.85 psu (Adkins et al., 2002).

The absolute magnitudes of cooling in the six surface boxes between MIS 5e and MIS 4 (Table B1) were chosen such that MOT decreased by 3.1 °C between MIS 5e and MIS 5a and by 0.9 °C across the MIS 5a–4 transition, consistent with the ice core MOT data (Shackleton et al., 2020, and this study). Because our MOT data provide no constraints on the spatial distribution of ocean temperature change, we make relatively simple assumptions on the spatial distribution of ocean cooling. The high-latitude Southern Ocean box temperature change from MIS 5e to MIS 4 (3.5 °C) is smaller than the other surface boxes (5.5 °C) so that the Southern Ocean box does not approach temperatures below the freezing point of seawater. The total relative change in modeled global mean ocean surface temperature is similar to the total relative change in a stack of 136 sediment core records (Table B1, Fig. 4; Kohfeld and Chase, 2017), although the absolute value of mean ocean surface temperature is lower in the model. As discussed in the main text, the choice in the spatial distribution of the modeled ocean temperature change may influence the strength of the modeled solubility pump (Khatiwala et al., 2019). Further experimentation should consider the interplay between different spatial patterns of ocean surface cooling, ocean circulation, and disequilibrium.

**Table B1.** Magnitude of sea surface cooling prescribed to the carbon cycle box model between 129 and 59.7 ka. The magnitude of modeled global mean ocean surface cooling is also given for the period from 126 TS2 to 59.7 ka to compare with published reconstructions (Kohfeld and Chase, 2017).

| Model box/Region | Temperature change from MIS 5e maximum to MIS 4 minimum |
|---|---|
| Low-latitude Atlantic | $-5.5\,°C$ |
| Low-latitude Pacific | $-5.5\,°C$ |
| Mid-latitude subantarctic | $-5.5\,°C$ |
| High-latitude Southern Ocean | $-3.5\,°C$ |
| High-latitude North Pacific | $-5.5\,°C$ |
| High-latitude North Atlantic | $-5.5\,°C$ |
| Model global mean ocean surface temperature | $-5.5\,°C$ |
| Global mean ocean surface temperature reconstruction (Kohfeld and Chase, 2017) | N/A TS3 (reconstruction begins at 126 ka with local temperature maximum at 124 ka) |
| | Temperature change from 124 ka to MIS 4 minimum |
| Model global mean ocean surface temperature | $-4.1\,°C$ |
| Global mean ocean surface temperature reconstruction (Kohfeld and Chase 2017) | $-3.9\,°C$ |

**Data availability.** Data presented in this study are available online at https://doi.org/10.15784/601415 (Shackleton, 2020).

**Author contributions.** JPS, EB, and VVP designed the research. SaS performed the noble gas measurements. JAM constructed the age model. SaS ran the MOT box model simulations. JAM ran carbon cycle model simulations. CB ran the WAIS Divide firn model simulations. MND led field logistics for Taylor Glacier sample acquisition. SaS, JAM, MND, and DB analyzed the data. SaS wrote the paper with input from all co-authors.

**Competing interests.** The authors declare that they have no conflict of interest.

**Disclaimer.** Publisher's note: Copernicus Publications remains neutral with regard to jurisdictional claims in published maps and institutional affiliations.

**Acknowledgements.** We thank Mike Jayred for drilling the core analyzed in this study and Kathy Schroeder for managing the Taylor Glacier field camp. The authors are also grateful to Thomas Bauska, Rachael Rhodes, Peter Sperlich, Isaac Vimont, Jake Ward, Heidi Roop, Peter Neff, Joe McConnell, Bernhard Bereiter, and Andrew Smith for their help with field logistics, drilling, and sampling of ice cores. Ice Drilling Design and Operations (IDDO) provided drilling support, and the US Antarctic Program provided logistical support for this project. Michael Bender is acknowledged for providing helpful feedback on early drafts of this paper.

**Financial support.** This research has been supported by the National Science Foundation (NSF) (grant nos. 1246148, SIO; 1245821, OSU; and 1245659, UR) and the NSF Graduate Research Fellowships Program (grant no. DGE-1650112). TS4

**Review statement.** This paper was edited by Alessio Rovere and reviewed by two anonymous referees.

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

## Remarks from the typesetter