# Peer review of "Evolution of mean ocean temperature in Marine Isotope Stage 4"

_Climate of the Past, 2021_

## Author Comment (AC1)

In your discussion of the solubility effect I suggest to consider new results from Khatiwala et al. (2019, doi: 10.1126/sciadv.aaw4981) that show disequilibrium effects lead to a much larger temperature effect on atmospheric CO2 that previously thought.

*Thanks for the suggestion. We have included discussion of the results of Khatiwala et al 2019 and their significance for our study in the updated manuscript:*

*Lines 224-234: Our data allow us to put new constraints on the role of the solubility pump in atmospheric $CO_2$ variations across the studied intervals. Out of the full ~80 ppm $CO_2$ from MIS 5e to MIS 4, our modelling suggests that MOT changes can explain 41±4 ppm. These estimates of the solubility pump agree well with the canonical 10 ppm/°C (Williams and Follows, 2011). However, our MOT data provide no information on the spatial distribution of ocean temperature change, which a recent study suggests plays an important role in modulating the strength of the solubility pump via changes in ocean saturation state (Khatiwala et al., 2019). The referenced study found that changes in air-sea disequilibrium between interglacial and glacial ocean conditions enhanced the solubility pump by ~60% during the last glacial maximum. If such disequilibrium effects are also relevant for the timescales and periods considered here, our solubility-driven estimates from the carbon cycle model simulations may be considered a lower bound. In particular, if the enhanced disequilibrium effect is linked to the onset of the glacial mode of ocean circulation at the MIS 5a-4 transition, then the solubility pump may play a larger role there than suggested from our simplified carbon cycle modeling.*

---

## Author Comment (AC2)

Shackleton et al. provide a reconstruction of mean ocean temperature (MOT) covering Marine Isotope Stage 4 (MIS 4, 74 to 59.5 ka BP) based on 56 new noble gas measurements performed on a shallow ice core from the Taylor Glacier blue ice area in Antarctica. Based on their new MOT reconstruction and previously published data covering the last and penultimate deglaciations the authors argue that most of the ocean cooling between the Last Interglacial (MIS 5e) towards the Last Glacial Maximum (MIS 2) occurred already across MIS 5, with little to no net change during MIS 3. The temporal resolution during MIS 4 is just high enough to allow the authors to speculate on millennial-scale MOT turning points and trends linked to the Atlantic Meridional Overturning Circulation (AMOC).

The authors place their data in a greater climatic context and discuss the potential links to other paleoclimatic parameters such as atmospheric CO2 and benthic d18O. With the help of a carbon cycle box model (Bauska et al. 2016), the authors provide a quantitative estimate of the effect of the cooling ocean on lowering the atmospheric CO2 concentrations over the course of MIS 5 and MIS 4. These are exciting new data and insights which add to a great body of literature seeking to understand climatic processes at play during the last glacial cycle.  Although some of the authors' interpretations concerning the millennial-scale variability are speculative at times and mostly leaning on previous findings for different climatic boundary conditions, I am convinced that the authors are mostly correct.

All authors of this study are well-recognized experts in ice core sciences. Among the authors are leading experts in the field of noble gas thermometry. The measurement of noble gas ratios is technically challenging and consumes large amounts of precious ice samples (>800g). Therefore, these data covering only a rather short time interval (MIS 4, ~15 ka) can still be considered substantial new material. I am convinced that this study will be of great interest to the paleoclimate community. I think the manuscript is well suited for publication in Climate of the Past after addressing some issues.

*We thank the anonymous reviewer for their thoughtful and overall positive feedback on the manuscript. They have included helpful suggestions and fair critiques of the interpretation of this new record, which are addressed below in the specific comments.*

**MOT in Fig. 1**

The authors compare their data to previously published records throughout the manuscript but do not show the data in their figures. While the data by Shackleton et al. 2020 appear in Fig. 3 in the context of a model-data comparison, the most recent data covering MIS 2 and the last deglaciation (e.g. Bereiter et al. 2018a) do not appear in any figure. I highly recommend adding previously published MOT records to Fig. 1. In this context, I feel the introduction would benefit from a brief summary of what we have learned from noble gas thermometry so far, referencing to Fig. 1. See comments below. Another argument for adding MOT to Fig. 1 is that the manuscript further compares benthic d18O to both old and new MOT reconstructions, yet there is no figure showing both data sets.

*This is a helpful recommendation. The MOT record has been be added to figure, and a brief summary of MOT findings has been added to the introduction:*

*Lines 50-54: High resolution reconstructions of MOT have been limited to the last two glacial terminations (Baggenstos et al., 2019; Bereiter et al., 2018a; Shackleton et al., 2019, 2020), but have*

*provided unique insight into the interplay of key climate variables. In addition to the long term warming across these deglaciations, millennial scale variations in MOT are observed, which are also seen in Antarctic isotope records (Masson-Delmotte et al., 2010), and correspond to changes in Atlantic Meridional Overturning Circulation (AMOC) (Mcmanus et al., 2004).*

**Point on full MIS 5-4 record**

I am sure it is not intentional, but, starting with the title, the authors imply multiple times that they present an extensive dataset covering the entire MIS 5-4 interval, lasting for about 60 ka. After all, the authors provide (only) data for MIS 4 (including a few thousand years during MIS 5a) covering 15 ka, leaving a gap with no MOT data of about 50 ka during MIS 5e to 5a. The authors do mention that data over MIS 5 are sparse, but I think that it is important to note that we don't know the evolution (as in trajectory) of the MOT during MIS 5e to 5a, all we can tell is that the net change between two endpoints.

My suggestions for the title are: "Evolution of mean ocean temperature in Marine Isotope Stage 4" or just "Mean ocean temperature during/across Marine Isotope Stage 4".

*Duly noted. The title has been revised to 'Evolution of mean ocean temperature in Marine Isotope Stage 4'*

**Millennial-scale variability**

Given the trends in the EDC dD record (Fig. 2b) I agree that the cooling may have happened in two stages at two different rates at the MIS 5a-4 boundary. However, looking at the data I am less convinced. How robust is this finding?

*This is a fair question. We include bootstrapping in our splined MOT record, which should help to remove any spurious trends, but a more rigorous test should be applied here. To test the robustness of the result, we divide the MOT data showing the MIS 5a-4 decrease into two sections, one defined by GI19 (shown in Figure 2 with gray shading) and the second from the onset of GS19 to 67.5 ka, where the MOT record reaches a minimum. We can then estimate the rate of MOT decrease and its uncertainty for each of these sections from the Monte Carlo simulations of the MOT data. We find a cooling of -0.41±0.09°C/kyr in the first stage of the 5a-4 transition, and -0.19±0.07°C/kyr in the second stage. A two-tailed z-test shows that the difference between these rates of MOT decrease is statistically significant (p=0.05).*

I agree with the authors that the AMOC (via the bipolar seesaw mechanism) is linked to MOT, which is a logical consequence of the strong correlation with the EDC dD record. However, I think the authors are jumping to conclusions. A direct link with the AMOC needs to be established with more than just one DO event (DO 19 at ~72 ka BP, Fig. 2) and also needs to involve direct AMOC reconstructions, which are available for this time period: Böhm et al. (2015) Strong and deep Atlantic meridional overturning circulation during the last glacial cycle, 517, 73-76, Nature, doi:10.1038/nature14059. However, given that MIS 4 is muted in its DO activity and the MOT data only allow a direct comparison with DO 19 I think adding this record would probably not add much at this stage. Delving any deeper into this topic would distract from the authors' main conclusions. I think it is OK to speculate that MOT and AMOC are linked even outside of glacial terminations; however, please clearly label the speculations as such.

*In response to the comments here we have toned down the language and emphasis on this point by reworking this section and combining it with the discussion on covariation of MOT and Antarctic temperature. We have removed language that implies a direct link between MOT and AMOC and instead discuss the data in the context of millennial scale climate variability:*

*Lines 275-277: However, our record demonstrates that Antarctic temperature and mean ocean temperature covary on millennial timescales during DO19, suggesting that their link is not unique to deglaciations, and is a general feature of the climate system.*

**Error analyses (section2.4)**

I am confused about the way the uncertainty (Fig 2.a) has been calculated. The authors mention that they have used a bootstrapping method (a resampling method, involving the exclusion of a certain number of data points for each run), but go on and say that this involved wiggling the data within their uncertainty boundaries (Monte Carlo technique). Please clarify if you indeed used a bootstrapping method or if this only involves the exclusion of the low data point at 62 ka BP described in the caption of Fig. 1. If you did use actual bootstrapping, please provide a little more detail on the method used.

*This was not clearly explained in this first draft of our manuscript. We have clarified in the updated manuscript that the bootstrapping technique is used to resample from the Monte Carlo simulations of the MOT time series to produce the splined record, but not to calculate errors associated with individual MOT datapoints:*

*Lines 127-130: To produce the splined MOT record, we use the 10,000 Monte Carlo iterations of the dataset, and randomly sample the 56 individual MOT points via bootstrapping using the Matlab function randsample with replacement. We then fit each of the 10,000 time series using a spline with a 2500-year cut-off period using the Matlab function csaps, and averaged the resulting splines to produce a final, smoothed version of our MOT record including uncertainty estimates.*

Looking at Fig. 2 I can see why the vertical error bars were omitted for clarity but I had to look at the data file to get an idea of the uncertainty associated with those measurements. I see that the propagated uncertainty of a single data ranges between 0.17 to 0.19°C, I think this should be mentioned in the text.

*Calculated uncertainty for individual samples (0.2°C) has been added to the text, as well as the pooled standard deviation of replicate samples (0.3°C, see below). However, we hesitate to report this uncertainty to the hundredth of °C, because it may give the reader a false sense of how precisely we know uncertainty in MOT, which does not include systematic errors (such as due to a change in ocean saturation state) that may exceed the analytical uncertainty. We have also included a note about systematic uncertainties in the section on error analysis:*

*Lines 125-127: The calculated uncertainty for an individual sample is 0.2°C (1$\sigma$), and the pooled standard deviation of the 11 replicate samples is 0.3°C. Systematic uncertainties (such as changes in ocean saturation state) may cause us to underestimate total uncertainty in our record.*

Out of 11 replicate measurements, 5 do not agree within their uncertainty boundaries. Does that mean that your uncertainty estimates are too low? Maybe I am missing something here?

*We appreciate the reviewer's careful evaluation of the data. For the 11 replicates, we find a pooled standard deviation of 0.34°C, which clearly exceeds the reported uncertainty estimates from Monte Carlo simulations of the data. However, the choice in replicates were not totally random. Eight were measured in a sampling campaign that occurred a year earlier than the rest of the samples and were remeasured to test for consistency between campaigns. The three others were selected because, upon initial review of the data, they appeared to be outliers in comparison to their nearest neighbors. If we don't include the 3 samples that were initially flagged as potential outliers and compare the 8 replicates that were done between two campaigns (which may represent a more randomly selected subset of the data) the pooled standard deviation is 0.20°C, which is more consistent with the reported uncertainty for individual points (0.17-0.19°C). While we believe the intentional remeasuring of potential outliers is a likely explanation for the higher uncertainties of replicate samples, we cannot be certain, so we have reported the pooled standard deviation of all 11 samples in the manuscript in addition to the estimated uncertainties from Monte Carlo simulations.*

I am confused about the spatial arrangement of replicate samples in the case of Taylor Glacier ice. How comparable are two replicates in your case? In an analogy to a classical ice core drilled on a dome or divide: Are we talking about true replicates from identical depth levels (but different cuts of the core) or quasi replicates from vertically directly neighboring ice from the same cut?

*The Taylor Glacier samples are large diameter (10"), so the replicates are from identical depths. Layers in this region of the glacier are horizontal, so replicates samples should be of the same age. This detail has been added to line 71.*

What are the greatest contributors to the final uncertainty? Is it the fractionation correction?

*As noted above, there may be systematic uncertainties that are not included that may cause us to underestimate the final uncertainty. As discussed in Appendix A, there are small, but systematic differences in MOT results when different approaches to fractionation corrections are applied. However, the analytical uncertainty of the noble gas ratios measurement is one of the largest contributors.*

**Box model(s) and d13C-CO2**

The carbon cycle model simulations performed within the scope of this study come as a surprise. In fact, the first time a box model is mentioned is only in section 2.3 where the box model parameterizations are described. It is slightly confusing that the authors use two different box models, one for the calculation of the MOT itself (Bereiter et al. 2018a) and a carbon cycle box model published by Bauska et al. (2016). The authors refer to both as the "box model". Do these models have names? Please make sure that the reader understands the difference. I only realized that there are two different models after having read the manuscript multiple times.

*In the updated manuscript, we have now systematically referred to them as the 'carbon cycle model' the 'MOT box model' to be sure there is not this same confusion for the reader. In the methods we have also added two individual sections (section 2.4: 'MOT calculations from corrected noble gas ratios' and section 2.5: 'Carbon cycle model calculations of the solubility pump'.*

The carbon cycle model is first introduced in the Discussion. I think this should be mentioned in the abstract including the most important results thereof.

*We have added the results from the box model to the abstract and description of the carbon model to methods:*

*Lines 20-21: Using a carbon cycle model to quantify the $CO_2$ solubility pump, we show that ocean cooling can explain most of the $CO_2$ drawdown (32±4 of 40 ppm) across MIS 5.*

*Lines 112-118: To estimate the effect of a cooling ocean on atmospheric $CO_2$ concentration via the solubility pump, we use a simple carbon cycle model (Bauska et al., 2016) to run a forward scenario of prescribed ocean temperature change from MOT constraints. The model consists of fourteen boxes representing the surface oceans (six boxes), intermediate oceans (two boxes), and deep oceans (three boxes), a well-mixed atmosphere (one box), and a terrestrial biosphere (two boxes). The model simulates thermohaline circulation and mixing, air-sea gas exchange, export production, sediment burial/ $CaCO_3$ compensation, and exchange of carbon between the atmosphere and terrestrial biosphere. More details on the model can be found in Appendix B.*

In this context, I am quite surprised that the authors chose not to show the d13C-CO2 model output of the carbon cycle model, given that the model used here (Bauska et al. 2016) was built for simulating the d13C-CO2 budget. This further makes me wonder why the d13C-CO2 record by Eggleston et al. 2016 is not discussed here. Eggleston et al. (2016) Evolution of the stable carbon isotope composition of atmospheric CO2 over the last glacial cycle, 2015PA002874, Paleoceanography, https://doi.org/10.1002/2015PA00287. The major feature in the record by Eggleston et al. is a large through in the last glacial cycle, with the strong drop in d13C-CO2 during MIS 4. I wonder if the authors can help to disentangle this feature?

*This is a good observation made by the reviewer, and a fair point. The reason that we have not included any discussion of the d13C results in the context of our new MOT data is that a new, high resolution d13C reconstruction was recently produced with the same ice core and will be submitted for publication in the near future. We hope the reviewer understands that we would prefer to leave a discussion of MOT constraints on d13C to a paper where it can be discussed in depth.*

Moving on to the conclusions, where to my surprise, the carbon cycle model results are mentioned first, although this is not the main conclusion of this study. I recommend reformulating the first paragraph of the conclusions and reorganize the conclusion section. Start by summarizing your measurement achievements first and state your main conclusion, that is that the MOT in MIS 4 and MIS 2 are the same (you don't mention this at all, despite it its importance in the abstract) followed by the evolution of MOT within MIS 4. Then add a summary of the consequences of your main findings and what you learned from the carbon cycle model. Avoid "clearly". Finally, move on to conclude on details and finish with your closing statement, which is great.

*We appreciate the reviewer's comments regarding the conclusions. We have substantially reworked the conclusion following these suggestions:*

*Lines 307-320:*

*Our record adds to the growing number of MOT reconstructions and provides unique insight into the climate conditions of MIS 4 within the context of the last glacial inception. The MOT record shows comparably cold and stable conditions in MIS 4 and MIS 2. As demonstrated in previous studies, MOT and Antarctic isotope records are remarkably correlated and covary on millennial timescales*

*during DO19, providing the first evidence of the connection between MOT, Antarctic temperature and inferred AMOC variability outside of deglaciations. Comparisons of coeval benthic $\delta^{18}O$, $CO_2$ and MOT show that while MOT reaches a minimum in the last glacial cycle by MIS 4, $CO_2$ and $\delta^{18}O$ don't achieve their glacial extrema until MIS 2; ocean cooling outpaced ice sheet growth and atmospheric $CO_2$ drawdown in the last glacial inception.*

*Using a carbon cycle model and ocean temperature constraints provided from our record, we demonstrate that ocean cooling played a major role in the early (MIS 5) stages of atmospheric $CO_2$ drawdown in the glacial inception (32 of ~40 ppm), a moderate role in the first half of the MIS 5a-4 transition (7 of ~20 ppm), a minor role in the second half of the 5a-4 transition (2 of ~20 ppm), and no measurable role in the ~20 ppm decrease between MIS 4 and MIS 2. This suggests an evolving control on atmospheric $CO_2$ during the glacial inception in which the solubility pump initially dominates but plays a progressively smaller role as the glacial inception progresses. MOT reconstruction of the entire MIS 5 interval would provide valuable insight into this apparent trend.*

**Data availability**

I commend the authors for providing an easily accessible and clearly structured data file online via https://www.usap-dc.org/view/dataset/601415 as linked to by a DOI link in the manuscript.

Please update the correct reference to this paper once accepted. The title given in the data file reads "Mean ocean temperatures achieve full glacial levels by Marine Isotope Stage 4 in the last glacial cycle".

*This will be updated if/when the manuscript has been accepted.*

**In-text citation style**

In cases where an author's name is part of the narrative the citation format remains the same as for indirect citations and does not change accordingly which is disrupting the reading flow. I recommend replacing e.g. "(Shackleton et al. 2019)" with "Shackleton et al. (2019)". Examples include: Menking et al. 2019 (line 70), Shackelton et al. 2019 (line 80), Pedro et al. 2018 (line 217), Sigman et al. 2010 (line 238)… etc.

*This is due to a bug in the reference software we're using, and we'll make sure that this is fixed in the final draft of the manuscript.*

**Line-by-line comments:**

Line 17-18: You also use data from Termination II, which is not part of the "last glacial cycle" in my understanding. Here a suggestion: … MOT reconstructions from the last and penultimate deglaciation… ?

*Done*

Line 18: …we find that the majority of the interglacial-glacial ocean cooling must have occurred across MIS 5

*This has been changed*

Line 18-19: what are "full glacial levels"? Please define "full glacial levels" and consider reporting previously published MOT of MIS2 or provide a measure of how those two MOT'S (MIS 4 vs MIS 2) differ

*This has been rephrased and a comparison to MIS 2 has been added:*

*Lines 17-19: Comparing this MOT reconstruction to previously published MOT reconstructions from the last and penultimate deglaciation, we find that the majority of interglacial-glacial ocean cooling must have occurred within MIS 5 and MOT reached equally cold conditions in MIS 4 as MIS 2 (-2.7±0.3°C relative to the Holocene, -0.1±0.3°C relative to MIS 2).*

Line 19-20: "Comparing MOT to…". This sentence is quite vague. Please reformulate and quantify the magnitude "CO2 drawdown" and "d18O increase". Please also mention that, based on box model experiments performed within the scope of this study, you estimate that up to XX% (instead of "most of") may be attributable to ocean cooling.

*This has been reformulated and specific values for CO2 d18O changes have been included:*

*Lines 20-22: Using a carbon cycle model to quantify the $CO_2$ solubility pump, we show that ocean cooling can explain most of the $CO_2$ drawdown (32±4 of 40 ppm) across MIS 5. Comparing MOT to contemporaneous records of benthic $\delta^{18}O$, we find that ocean cooling can also explain the majority of the $\delta^{18}O$ increase across MIS 5 (0.7 of 1.3‰).*

Line 21: I think you need to explain what you mean by "millennial scale climate variability". Which hemisphere are you referring to? Is the climate variability really setting the MOT?

*Fair point. This has been removed and replaced with 'The timing of ocean warming and cooling in our record, and comparison to coeval Antarctic isotope data suggest an intimate link between ocean heat content, high Southern latitude climate, and ocean circulation on orbital and millennial timescales'*

Line 26: Please introduce "ka BP"

*Done.*

Line 26: Please mention that your MIS definition is following Lisiecki and Raymo (2005).

*We have added a citation to Lisiecki and Raymo, 2005 here.*

Line 27: Please provide a (layperson's) definition of your definition of the "glacial inception

*This has been added.*

*Lines 27-28: However, the glacial inception - the transition from interglacial to glacial maximum – also…*

Line 47-49: Second, the comparison of MOT… Where is that comparison? Please add MOT data to Fig. 1.

*MOT data have been added to Fig 1.*

Line 49-56: "Third,…" I am sure it is not intentional, but this section sounds like you can say much about MOT changes associated with several DO events occurring during MIS 5d to MIS 5a. However, your data covers only one DO event with high-resolution MOT data (which is absolutely great, no need to overstate the extent of your study). I also think the introduction of DO events at this point in the manuscript disrupts the flow of reading at the end of the introduction where you motivate this study. I suggest adding a short paragraph just before "Here we reconstruct MOT from…" introducing important records relevant to this study and summarizing what we have learned from MOT reconstructions so far, including the suggested link between AMOC and MOT. This new paragraph can then be followed by your existing paragraph (starting in a new line) outlining the four purposes, i.e. what your study adds to this body of literature and why this is important.

*We've specified that we only have MOT data for DO 19, and have combined this suggestion and the reviewer's previous suggestion about adding a brief summary on what we've learned from noble gas thermometry thus far:*

*Lines 52-57: In addition to the long term warming across these deglaciations, millennial scale variations in MOT are observed, which are also seen in Antarctic isotope records (Masson-Delmotte et al., 2010), and correspond to changes in Atlantic Meridional Overturning Circulation (AMOC) (Mcmanus et al., 2004). These deglacial features of MOT suggest an intriguing link between ocean circulation and ocean heat content. However, it is unclear if this link is unique to terminations or also applies to DO events (Dansgaard et al., 1982), millennial-scale climate oscillations that are thought to be linked to AMOC variability within glacial intervals (Lynch-Stieglitz, 2017; Stocker and Johnsen, 2003).*

Line 55: Please rephrase. Our record provides important, yet limited, insight that helps understand … by providing high-resolution MOT data over DO event 19 which is marking the MIS 5a/4 transition. Or likewise.

*This has been rephrased to better reflect the limitation in scope:*

*Lines 62-63: Third, it allows us to test if the link between changes in ocean circulation and heat content exists during DO event 19 at 72.1 ka.*

Line 56: mention your carbon cycle model here

*Carbon cycle model and reference has been added:*

*Lines 63-64: Last, using a simple carbon cycle model (Bauska et al., 2016) we estimate the contribution of whole-ocean cooling to the decrease in atmospheric $CO_2$ across MIS 5 and the MIS 5a-4 boundary due to the solubility pump.*

Line 57: atmospheric CO2

*Done.*

Line 60: change "20 meters" to "20 m" for consistency

*Changed.*

Line 64: I believe you measured 11 replicates (see data file)

*Changed.*

Line 64: What was the average sample weight? Mean temporal resolution?

*These details (806 g and 330 year resolution) have been added (see line 72).*

Line 77: The title of this section is confusing. See the comment above on the box model conundrum. I recommend replacing "box model parameterizations" with "MOT calculation" or similar.

*We have moved the with the details of the box model to a separate section and renamed it to 'MOT calculation from corrected noble gas ratios' for clarity.*

Line 79-83: Please add a sentence explaining why the use of isotope ratios inert gases is superior to the firn model-based correction for thermal fractionation (Bereiter et al. 2018a). Which gases have been measured for the purposes of this correction in this study? Please add this information to section 2.1.

*Here, we wouldn't necessarily argue that the use of inert gas isotope ratios to correct for thermal fractionation is superior to the firn-based modeling approach. In the case of WAIS Divide, previously published model output for the firn thermal gradient was readily available, which is why it was used in Bereiter, 2018. However, this is not the case for Taylor Glacier. We've added these details to Appendix A and have added the details on which inert gas isotope ratios were measured (see line 90).*

*Lines 329-334 (in Appendix A): The approach of fractionation correction in this study differs slightly from that of (Bereiter et al., 2018a), which uses previously published firn model output from (Buizert et al., 2015) to correct $Kr/N_2$, $Xe/N_2$ and Xe/Kr for thermal fractionation. To correct $Kr/N_2$, $Xe/N_2$ and Xe/Kr for gravitational fractionation, the (Bereiter et al., 2018a) study uses measured $\delta^{40/36}Ar$, which is first corrected for thermal fractionation using this same firn model output. While in the case of the WAIS Divide ice core, previously published firn model output was readily available for this purpose, no such model output exists for Taylor Glacier, which is why this method of fractionation correction was not considered in this study.*

Line 83: Where does the Argon suddenly come from? See comment above.

*We have added the isotope ratios used (see line 90) and have moved this last sentence to the appendix, since it's a very minor (<0.005 per mil) correction.*

Line 78-83: How large is that correction? Please quantify in both absolute and relative terms.

*We have added the following to Appendix A (Lines 356-364):*

*The average magnitude of the fractionation corrections for $\delta Kr/N_2$, $\delta Xe/N_2$ and $\delta Xe/Kr$ respectively were 5.0‰, 8.9‰, and 3.9‰ for the Taylor Glacier samples in this study. The fractionation-corrected, atmospheric values found for $\delta Kr/N_2$, $\delta Xe/N_2$ and $\delta Xe/Kr$ are -1.4‰, -3.6‰, and -2.1‰ respectively. While the fractionation corrections are large relative to the atmospheric signal, they are dominated by gravitational fractionation, which is physically well understood and constrained by the measured isotope ratios. For context, the average magnitude of fractionation correction for the WAIS Divide samples in this study are 22.7‰, 42.5‰, and 19.3‰ for $\delta Kr/N_2$, $\delta Xe/N_2$ and $\delta Xe/Kr$. The large difference in the magnitude of the applied fractionation correction between ice cores, but good agreement in corrected noble gas ratios and resulting MOT gives us some confidence in our ability to apply these corrections.*

Line 89-96: Maybe I am mistaken here, but I think this section belongs into the Discussion. Maybe part of it can go into the new introductory section requested above?

*Upon a closer look at the MIS 2 EDC data, we realized that when we were considering MOT variability within MIS 2, we had included EDC data from within the bubble-clathrate transition zone, which has been shown to cause potential problems with MOT reconstructions (see Bereiter et al, 2018). Without this data, the standard deviation of MIS 2 samples is 0.3°C. While this may be slightly larger than the MOT variability within the Holocene for this core, it is still less than the reported analytical uncertainty for individual MOT samples in this study (0.4°C). We therefore do not believe the evidence for greater MOT variability within MIS 2 (compared to the Holocene) is so compelling and have revised this statement and have moved some of it to the results:*

*Lines 94-99: For the Taylor Glacier samples, we compare our data to five early Holocene (10.6 ka) replicate Taylor Glacier samples from (Shackleton et al., 2020). WAIS Divide samples are reported relative to the average of Holocene samples from 11-10 ka (n=4) (Bereiter et al., 2018a). WAIS Divide (Bereiter et al., 2018a) and EPICA Dome C (EDC) (Baggenstos et al., 2019) records both suggest that the entire Holocene was a very stable interval for MOT (1$\sigma$ standard deviations of 0.2°C and 0.1°C respectively for all Holocene samples), so the particular choice in reference Holocene interval has minimal impact on the reported results.*

*Lines 143-147: For WAIS Divide samples, we compare the measured MIS 4 samples to all available, previously published (Bereiter et al., 2018a) MOT data from MIS 2 (24 – 18 ka) with applied the fractionation corrections and MOT box model parameterizations used in this study. The difference in WAIS Divide MOT results for the full MIS 2 interval (n=11) versus 20-19 ka (n=4) differ by less than 0.01°C, so the difference in the selected intervals to define MIS 2 for each core should not affect the MIS 4-2 comparison.*

Line 123: Any idea why WAIS shows such large scatter? Is the offset relative to Taylor Glacier significant? This ice is very far away from the bubble/clathrate transitions zone. I realize that these samples are only about 60 m above the bedrock at WAIS. Any idea how this could have an impact on the noble gas ratios?

*This is a good question, but with the few samples from the WAIS Divide core it's difficult to assess. The scatter is reduced if the Bereiter, 2018 correction is applied, so it's likely related to noise in the inert isotope data (used to correct for thermal fractionation), rather than the noble gas ratios.*

Line 126: …WAIS Divide is 0.2°C warmer…

*Changed.*

Line 130: …substantial MOT warming towards the end …

*Changed.*

line 145: replace "considerable portion" with an actual estimate, 23%?

*This has been placed with 'roughly one third'.*

Line 146: remove "clearly" or replace with "may" or similar

*Changed to 'trends appear distinct in their shape'.*

Line 147: "It is notable that MOT was already low during MIS 5a (Fig.3)." Unclear. Where is MIS 5a in Fig 3?

*This section has been reworked (and this sentence was removed) but we have added dates to clarify the timing of the intervals that we are referencing and have labelled them in Fig 3 (now Fig 4).*

Line 154: Benthic d18O records changes in deep seawater d18O and is used as a proxy for deep-water temperature.

*Following the comments of reviewer 2, much of this section (including line 154) has been removed.*

Line 160: "-71 m". Relative to what?

*Following the comments of reviewer 2, much of this section (including line 160) has been removed.*

Line 179: Swap AA cooling and MOT decrease

*Following the comments of this reviewer and reviewer 2, this section was substantially reworked and line 179 has been removed.*

Line 179-182. Long sentence that is hard to read with four "ands" in it. Please split up and reformulate.

*Following the comments of this reviewer and reviewer 2, this section was substantially reworked line 179-182 has been removed.*

Line 187-191: Ok, but speculative. I recommend acknowledging that by starting the sentence with: We speculate…

*Following the comments of this reviewer and reviewer 2, this section was substantially reworked line 187-191 has been removed.*

Line 203 onwards: Please provide R² values to underline your statement of strong correlation.

*We have included $r^2$ values for the all available MOT data ($r^2 = 0.94$) and for the new MOT data set ($r^2 = 0.59$) to this section.*

Line 225: add "to", not to be

*Done.*

Line 228-229: add "in model simulations" to the end of the sentence. Or similar.

*Done.*

Line 245: The benthic δ18O record is only shown in Fig. 1, but not Fig. 3.

*This was an error, we meant to refer to Fig. 4 (which is now Fig 3).*

Line 250: remove parentheses

*Done.*

Line 252-255: Mention Fig. 4c to make sense of the color code.

*Thanks for catching this. Done.*

Line 271 onwards: "Our record provides the first observational evidence that MOT responds to AMOC changes outside of deglaciations…" I agree with what you say but would recommend toning this down. For example: This is the first indication that AMOC changes may also be associated with MOT during the last glacial and not only during deglaciations. We speculate that a similar pattern can be expected for MIS 3 where DO events are more frequent… Do you?

*Fair point. We have toned down the language of this section:*

*Lines 309-311: As demonstrated in previous studies, MOT and Antarctic isotope records are remarkably correlated and covary on millennial timescales during DO19, providing the first evidence of the connection between MOT, Antarctic temperature and inferred AMOC variability outside of deglaciations.*

Line 282: replace "fuller" with "better" or "improved"

*Replaced with improved.*

Line 326: Please quantify these systematic differences.

*Done.*

*Lines 370-374: Thus, the systematic differences in MOT results using different fractionation correction methods, or between the three noble gas ratios are largest (up to 0.4±0.3°C between individual noble gas ratios and 0.3±0.3°C between correction methods) when comparing results between glacial and interglacial intervals but are minimal (up to 0.1±0.3°C between correction methods and <0.01±0.3°C between noble gas ratios) when comparing MOT results between the MIS 4 and MIS 2 glacial intervals.*

Line 341: replace "grasp" with "understanding"

*Done.*

Line 362-363: repetition of statements in the main text.

*Removed.*

Line 364-366: Better move this sentence to the main text?

*This has been moved and we have included additional discussion in the main text (following comments from Andreas Schmittner).*

**Figure 1:**

- Add new and previously published MOT data in new panel
  - *Done*
- Allow for a little more space in the vertical direction to help with the y-axes (see below)
  - *Done*
- Make sure the respective y-axes covers the full range of data shown, for example, the lowest value on the CO2 axis is 200 ppm but the data goes down to 180 ppm
  - *Done*
- Add major ticks to x-axis (time axis)
  - *Done*
- Add reference for insolation data in panel g)
  - *Done*

**Figure 2:**

- Consider adding error bars to panel a)
  - *We tested this out and it made the figure look bit too busy, so we decided not to include them*
- Add major ticks to x-axis (time axis)
  - *Done*
- Line 546: 62 ka BP

- *We decided to remove the dotted line (and this statement), because bootstrapping should, in theory, account for outliers and retrospectively including this spline feels a bit like cherry-picking.*

**Figure 3:**

- Add labels for panels a), b), c)
  - *Done*
- Add major ticks to both axes
  - *We might have different definitions of ticks here, but major ticks are already on these axes*
- Line554: refer to Appendix B for the dD seawater correction
  - *Done*

**Figure A1 and A2:**

- Add labels for panels a), b), c)
  - *Done*
- Add major ticks to x-axis (time axis)
  - *Done*
- Adjust length of y-axes to cover the full range of data but keep them as short as possible.
  - *Done*
- It is hard to differentiate between the different colours of x's. Maybe different symbols for the different types of corrections would be helpful here? You could use open symbols for WAIS and full symbols for TG?
  - *Good suggestion, done.*

---

## Author Comment (AC3)

This paper presents a new MOT reconstruction from noble gas measurements between 74 and 59.5 ka. These new data are combined to other MOT data obtained over the last climatic cycle. They are compared to benthic d18O and Antarctic dD and used to discuss the effect of solubility pump.

As all previous studies showing MOT reconstructions based on noble gas measurements, these data represent a huge analytical effort and they are worth to be published because they will be very useful. The discussion of the data could however be improved in a future version of the manuscript. The conclusions conveyed by the abstract are fine and the discussion and conclusion of the manuscript should thus be reorganized to be in line with the abstract.

List of comments:

What is the exact scientific aim of the paper ?

- Make the link between CO2 atmospheric concentration and MOT ?
- Discuss the link between MOT and Antarctic temperature ?
- Compare the MOT between MIS 2 and MIS 4 ?
- Discuss the MOT dynamic over millennial-scale DO events ?
- Separate sea level and deep ocean temperature contribution in benthic d18O stack ?

*We appreciate the anonymous reviewer's overall positive comments on the manuscript and their clear and constructive criticism on how to better convey the main arguments of the paper. The scientific aim of the paper is to present new MOT reconstruction for MIS 4 and the MIS5-4 transition and discuss the climatic implications of these data for our understanding of atmospheric CO2 and climate change. Based on the reviewer's comments we have substantially restructured the discussion and conclusion sections and address specific comments below.*

Figure 1: I find it confusing to have identification of MIS4 and MIS2 through intervals between vertical dashed line and black bars of different width to define MIS 4 and MIS 2 MOT -> Better find another definition for the black bars like "cold MIS 4 MOT" + better explain how these black bars were defined.

*In this first draft of the manuscript we did not adequately explain the reasoning behind the different intervals for MIS4/MIS2 comparison. The MIS 2 (or LGM) interval is from the previously published Bereiter et al study, which did not extend through all of MIS 2 (as defined by benthic d18O). For our MIS 4 record, we were concerned about misalignment between ice core and sediment records to define MIS 4, so chose the interval of low atmospheric $CO_2$ / EDC dD to define MIS 4 in this manuscript. We have specified this in the updated manuscript:*

*Line 140-147: Here, we do not use the intervals identified and defined by benthic $\delta^{18}O$ to compare MOT in MIS 4 and MIS 2, as the alignment of ice core and sediment records is uncertain, particularly in MIS 4. Instead, we define MIS 4 as the interval in which $CO_2$ and Antarctic temperature remain low and stable (70.3-63.7 ka, or Greenland Stadial 19 and Interstadial 18). For Taylor Glacier samples, we compare MIS 4 samples to five replicate MOT samples from MIS 2 (19.9 ka). For WAIS Divide samples, we compare the measured MIS 4 samples to all available, previously*

*published (Bereiter et al., 2018a) MOT data from MIS 2 (24 – 18 ka), but applying the fractionation correction used in this study. The difference in WAIS Divide MOT results for the full MIS 2 interval (n=11) versus 20-19 ka (n=4) differ by less than 0.01°C, so the difference in the selected intervals to define MIS 2 for each core should not affect the MIS 4-2 comparison.*

Figure 2: The MOT temperature increase between 64 and 60 ka is not discussed in the manuscript while it seems that a strong MOT increases occurs between 62 and 60 ka while the EDC dD increase is less marked than between 64 and 62 ka when the MOT is stable. It could be argued that there are not enough MOT points and some scattering but this is equivalent to the period between 70 and 68 ka which is discussed in the text as the second phase of MOT decrease during MIS 4.

*This is a fair point, and a related question about the MOT trend between 70-68 ka was raised by reviewer 1. To evaluate whether the apparent decoupling between dD and MOT during GS18 is statistically significant (or, instead, may be attributed to scatter) we compare the correlation between dD that has been smoothed to remove high frequency variability (see figure 4 caption for details) to contemporaneous MOT data ($r^2=0.57$). This correlation with dD is lower than what is found when comparing all available MOT data to contemporaneous dD ($r^2=0.94$, figure 4a). However, the MOT range for this subset of the data is relatively narrow, so the lower signal to noise ratio may reduce the expected correlation. Based on the estimated uncertainty of individual MOT data from the pooled standard deviation of replicate samples in this record (0.34°C), we can predict the expected correlation between MOT and contemporaneous dD if we assume that the true dD and MOT signals are perfectly correlated ($r^2=1$), and that the lower correlation is due entirely to noise in the MOT data. Based on this assumption, we would predict an $r^2$ of 0.44±0.20, compared to the actual correlation $r^2=0.57$. This would suggest that the apparent decoupling between MOT and dD during this interval may be due to random noise.*

How robust is the MOT increase during GS 20 ? If we consider only the GS 20 data points (I.e. do not consider the two GI 20 data points), there is no MOT tendency over GS 20.

*Using a one tailed z-test on our Monte Carlo simulations of the data and including the two MOT data points that mark the low at the end of GI20, we find a statistically significant (p=0.03) increase in MOT at the onset of the record (during GS20). However, without the two low points at the end of GI20, the MOT increase during GS20 is not robust (p=0.48). While we acknowledge that the analytical uncertainties of our record present a challenge in detecting the finer scale variability shown in our record, we respectfully push back on the reviewer's comment here and argue that the noteworthy aspect of this early part of our MOT reconstruction is that MOT is increasing at all, given that this interval is widely regarded as a period of long-term cooling.*

Except for the GI 19 evolution, there is not so strong evidence for a fine scale correlation between dD and MOT on this figure.

*We respectfully disagree with the reviewer's comment here. There is a statistically significant correlation between dD and MOT for this record ($r^2=0.59$), which is lower than the correlation of all available MOT data versus dD ($r^2=0.94$). However, as mentioned above, the signal to noise ratio for this record should be lower than that of the (previously published) terminations. Using the pooled standard deviation of replicate samples from this study as a predictor of random noise (as above), we would predict an $r^2$ of 0.58±0.09 for the record published here and $r^2 = 0.93±0.01$ for all available*

*MOT data versus dD if MOT and dD were perfectly correlated and any lower correlation is due to random noise. We have added this point to section 4.1.3:*

*Lines 241-249: As highlighted in this, and several other MOT studies (Bereiter et al., 2018a; Shackleton et al., 2019, 2020), one of the most striking features of MOT records is their strong correlation to Antarctic water isotope records (Fig. 4a). For the MOT data from this study, we find a lower correlation between MOT and EDC $\delta^2H$ (n=56, $r^2$ = 0.59) than between all available MOT records (n=243, $r^2$ = 0.94). However, MOT and $\delta^2H$ data for this interval cover a relatively narrow range compared to other records, resulting in a lower signal to noise ratio, and thus may explain the lower correlation. To test this hypothesis, we use the pooled standard deviation of replicate MOT samples (0.3°C) as a predictor of random noise in the MOT record to estimate the expected correlation between $\delta^2H$ and MOT if we assume they are perfectly correlated ($r^2=1$). Under these assumptions, we would predict $r^2$ values of 0.58±0.09 and 0.93±0.01 for the MIS 4 subsample and all MOT samples respectively, which is consistent with the observed values.*

Figure 3: Following last comment, I am not confident that the Model MOT can be drawn as shown on the bottom panel with details at a scale of a few ka. Without more MOT data between 120 and 75 ka, and especially over the 120 – 110 ka strong modelled MOT decrease and large MOT increase and decrease between 88 and 78 ka, the modelled evolution is not robust which casts doubt on the interpretation in term of CO2 solubility pump between MIS 5d and MIS 5a.

*We agree with the reviewer that the modelled results within the region of ~120-75 ka (where there is a gap in MOT data) should not be overinterpreted. The purpose of the carbon box model was to demonstrate that the change in CO2 across (but not within) MIS5 could be mostly explained by ocean cooling. We attempted to show this with the arrows pointing to the start and end of MIS5 (where we do have MOT data) and text in the figure with the net change in MOT and modelled CO2 over MIS5. However, we did include some speculation about the $CO_2$ variability within the gap in MOT data at the end of the figure 3 caption, which we have removed and replaced with the statement 'model results within 120-74 ka should be interpreted with caution, as MOT data do not exist for validation'. We have also included a caution about this in the main text:*

*Lines 201-203: We emphasize that the available MOT data spans 9 kyr at the onset and 2 kyr at the end of the long (~57 kyr) MIS 5 interval, so our insight into the role of the solubility pump on $CO_2$ variations within MIS 5 is limited.*

p.5, l. 142-146: it is difficult to understand what is described here. It would help to clearly give the period (with dates) that you are discussing here. + the evolution after 70.5 ka is not very clear due to the lack of MOT data and scattering.

*This is a fair point, which was also brought up by Reviewer 1.  We have added the specific periods with dates that to the manuscript, which are defined by GI19 (72.1-70.3 ka) and the second from the onset of GS19 to 67.5 ka, where the MOT record reaches a minimum. To test if the rates of MOT decrease are robustly different, we can estimate the rate of MOT decrease and its uncertainty for each of these intervals from the Monte Carlo simulations of the MOT data. We find a cooling of -0.41±0.09°C/kyr in the first stage of the 5a-4 transition, and -0.19±0.07°C/kyr in the second stage. A two-tailed z-test shows that the difference between these rates of MOT decrease is statistically significant (p=0.05).*

Section 4.2 is confusing while it is a good idea to use MOT to decipher sea level contributions from deep ocean temperature on benthic d18O. What is the purpose of this section? Quantify the uncertainties in the reconstruction of MOT through sensibility to sea level value? If so, it is probably better in the annex or in the result section? then you again discuss the link with benthic d18O and sea level in section 4.4. The flow of ideas is difficult to follow.

*We appreciate the reviewer's comments/suggestion. In the updated manuscript, we have removed most of this section and combined it with the discussion on trends in coeval MOT and d18O.*

Sections 4.3 and 4.4 present the link between Antarctic temperature and MOT and invoke change in AMOC. The discussion on the link between Antarctic temperature and MOT should be gathered in a unique section for an easier reading of the manuscript.

*Thanks for this suggestion – we have followed it and combined these sections.*

The end of section 4.3 focuses on the temperature and ice volume of MIS2 vs MIS 4 which is quite disconnected from the beginning of the section. Try to reorganize the full discussion to convey clear conclusions and messages.

*We have moved the end of this section into a relatively brief separate section (now section 4.2: The cold and stable MIS 4 interval), which also includes a discussion on why temperatures in MIS 2 and 4 may be comparable. This new section is admittedly speculative, but we label it as such.*

It seems that you want to discuss:

- the MOT at MIS 4 compared to MOT during MIS 5e and MIS 2 with implication on the CO2 atmospheric concentration
- The link between MOT and Antarctic dD at glacial – interglacial and millennial scale with a discussion on the associated mechanisms

The discussion on d18Obenthic is not very clear here – Is it a perspective of this study to compare with d18Obenthic or should these data be used to refine uncertainty in the MOT determination.

*The discussion has been substantially reworked to reflect our main messages. The main sections now include 4.1.1: Evolving control of ocean temperature and ice sheet volume on benthic $\delta^{18}O$, 4.1.2 Early role of ocean cooling in atmospheric $CO_2$ drawdown, and 4.1.3 Strong correlation between MOT and Antarctic climate on orbital and millennial timescales.*

Conclusions:

- to be rewritten (the abstract is more explicit)
  - *Following the restructuring/revising of the discussion we have rewritten the conclusions to be more consistent with the main arguments of the manuscript.*

- the discussion on MIS 4-3 beginning on l. 266 was not present (or I missed it) in the sections of the discussion.
  - *This has been removed*
- The paragraph beginning l. 276 seems disconnected.
  - *The conclusions have been reorganized so that this paragraph immediately follows the discussion on the value of complete MOT records within MIS 5.*